# TODO: Enhancing LLM Alignment with Ternary Preferences

**Yuxiang Guo**[1,2*]   **Lu Yin**[1,3†]   **Bo Jiang**[2]   **Jiaqi Zhang**[1‡]
[1]Meituan Inc.   [2]Beihang University   [3]University of Surrey
{irisg,jiangbo}@buaa.edu.cn
l.yin@surrey.ac.uk   zhangjiaqi39@meituan.com

## Abstract

Aligning large language models (LLMs) with human intent is critical for enhancing their performance across a variety of tasks. Standard alignment techniques, such as Direct Preference Optimization (DPO), often rely on the binary Bradley-Terry (BT) model, which can struggle to capture the complexities of human preferences—particularly in the presence of noisy or inconsistent labels and frequent ties. To address these limitations, we introduce the **T**ie-rank **O**riented **B**radley-**T**erry model (TOBT), an extension of the BT model that explicitly incorporates ties, enabling more nuanced preference representation. Building on this, we propose **T**ie-rank **O**riented **D**irect Preference **O**ptimization (TODO), a novel alignment algorithm that leverages TOBT's ternary ranking system to improve preference alignment. In evaluations on Mistral-7B and Llama 3-8B models, TODO consistently outperforms DPO in modeling preferences across both in-distribution and out-of-distribution datasets. Additional assessments using MT Bench and benchmarks such as Piqa, ARC-c, and MMLU further demonstrate TODO's superior alignment performance. Notably, TODO also shows strong results in binary preference alignment, highlighting its versatility and potential for broader integration into LLM alignment. The implementation details and datasets can be found in `https://github.com/XXares/TODO`.

## 1 Introduction

Large language models (LLMs) demonstrate remarkable potential in various tasks (Huang et al., 2021; Hendrycks et al., 2021; Shi et al., 2023), with performance gains linked to better alignment with human intent (Mishra et al., 2022; Christiano et al., 2023; Wu et al., 2023). The alignment process typically involves two stages: Supervised Fine-Tuning (SFT) to establish instruction following abilities (Thoppilan et al., 2022; Sanh et al., 2022; Mishra et al., 2022), followed by preference fine-tuning to refine the model's alignment with human preferences (Ziegler et al., 2020; Christiano et al., 2023). This stage typically employs either reinforcement learning (RL)-based (Schulman et al.; OpenAI, 2023; Ramamurthy et al., 2023) or RL-free methods (Rafailov et al.; Azar et al.; Saeidi et al.; Meng et al., 2024), both leveraging the preference datasets. Effective alignment is enhanced by the diversity of training data, enabling LLMs to accurately learn from high-quality pairwise responses (Cui et al., 2023; Song et al., 2024; Saeidi et al.).

Current alignment methods relying on the Bradley-Terry (BT) (Bradley & Terry, 1952) model consider only two preference rankings: preference and dis-preference, which restricts the diversity of learnable information. A notable challenge is the inconsistent quality of the pairwise preference data, often showing minimal discernible differences (Nvidia et al., 2024; Amini et al., 2024). Table 1 shows a sample from the Ultrafeedback-binaried dataset[1] (Tunstall et al.), which is commonly used in preference alignment procedures (Tunstall et al.; Hong et al.). In this example, both responses have identical quality score of 8.5 from GPT-4 evaluations (OpenAI, 2023), differing only

---

[*]Work done during the internship at Meituan Inc.

[†]Work done during the research at Meituan Inc.

[‡]Corresponding author.

[1]`https://huggingface.co/datasets/HuggingFaceH4/ultrafeedback_binarized`

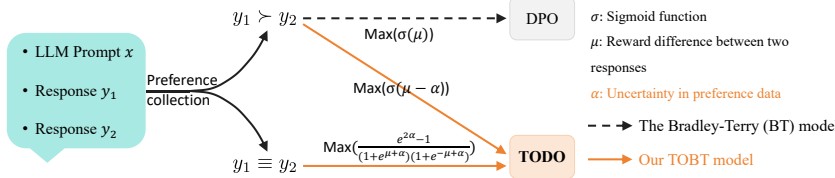

Figure 1: Comparison of DPO and TODO. DPO relies on the BT model, which is only capable of handling binary preferences. When responses are tied, it either learns incorrect preference information or discards tied data, preventing learning from such data. In contrast, the proposed TODO model can directly model ternary preferences. Based on this, TODO can learn more information from tied data and exhibits better robustness against potential noise in binary preference data.

in narrative sequence and text format. In practice, we observe a considerable amount of tie data in common preference datasets and chat arenas judged by humans, as detailed in Appendix A.1. These ties encompass a variety of information, necessitating a more nuanced analysis. However, existing preference optimization techniques, such as Direct Preference Optimization (DPO) (Rafailov et al.), are constrained by their reliance on the binary BT model and struggle to effectively manage tie relations. How to learn useful information from tie data and achieve nuanced preference modeling in the alignment process remains to be explored, which is the goal of this paper.

| Prompt | Q: Ninette de Valois was involved professionally in what activity?\nA: |
|---|---|
| R1 | Ninette de Valois was professionally involved in ballet as a dancer, choreographer, and founder of the Royal Ballet in London.\nConfidence: 95% (**score: 8.5**) |
| R2 | Ninette de Valois was involved professionally in the field of ballet. She was a choreographer, dancer, and founder of the Royal Ballet in London. (**score: 8.5**) |

Table 1: One pair of responses in Ultrafeedback-binaried Dataset which have identical quality scores evaluated by GPT-4. R1 is treated as the preferred response and R2 is treated as the dispreferred one.

Our primary contributions can be unfolded as:

❶ We enhance the existing human preference alignment process by incorporating a "tie" ranking, transcending the traditional binary rankings as depicted in Figure 1. We first extend the BT model into the **T**ie-rank **O**riented **BT** (TOBT) model. The TOBT model incorporates the concept of preference uncertainty, allowing for the representation of ties alongside "prefer" and "disprefer" rankings. This innovation enables a more comprehensive handling of preference relations.

❷ Building on the TOBT model, we introduce the **T**ie-rank **O**riented **D**irect Preference **O**ptimization (TODO) algorithm. TODO is designed to accommodate ternary ranking relations, offering a nuanced approach to preference alignment. By integrating the tie relation, TODO is capable of learning from a broader spectrum of preference information, enhancing the adaptability and accuracy of LLMs.

❸ We use Mistral-7B and Llama 3-8B to conduct experimental validation. First, we evaluate the effectiveness of DPO and TODO in terms of preference modeling accuracy. Our evaluation spans both in-distribution dataset, drawn from the same source as the training data, and out-of-distribution dataset, notably the Reward Bench (Lambert et al.). Results indicate superior preference modeling by TODO. Additional assessments on MT Bench (Zheng et al., 2023) and other popular benchmarks such as Piqa (Bisk et al., 2019), ARC-c, ARC-e (Clark et al., 2018), Hellaswag (Zellers et al., 2019), MMLU (Hendrycks et al., 2021) and Winogrande (Sakaguchi et al., 2019) confirms TODO's enhanced alignment capabilities. Finally, we provide an intuitive analysis highlighting TODO's advantages over DPO in two dimensions: enhanced granularity in preference modeling and increased diversity in the acquired information.

❹ TODO can also be directly applied in *binary* preference alignment process, outperforming DPO with standard binary preference datasets. Furthermore, the proposed TOBT model can be utilized not only in offline policies like DPO but also can be integrated into other online policies or used to train a reward model.

## 2 RELATED WORKS

**Preference alignment in LLMs.** Current methods for aligning preferences in LLMs often utilize the BT model (Bradley & Terry, 1952) and the Plackett-Luce ranking model (Plackett, 1975) to capture human preferences. RL-based approaches, such as Reinforcement Learning from Human Feedback (RLHF, (Schulman et al.)), require a reward model to assess generated responses, which is typically trained on pairwise preference data gathered from crowdworkers (Wang et al., 2024) or by utilizing another LLM as an evaluative judge (Bai et al., 2022; Cui et al., 2023). RL-free algorithms, such as Direct Preference Optimization (DPO) (Rafailov et al.) and its variants (Azar et al.; Ethayarajh et al., 2024), can directly optimize human preferences by introducing an implicit reward. DPO offers a stable and computationally lightweight method to align with human intent. To address the potential overfitting problem in DPO, Identity Preference Optimization (IPO) (Azar et al.) and Reward-aware Preference (RPO) (Nvidia et al., 2024) has been proposed by introducing differences between pairwise responses. Furthermore, Kahneman-Tversky Optimization (KTO) (Ethayarajh et al., 2024), was proposed by directly maximizing the utility of generations instead of maximizing the log-likelihood of preferences, and Hong et al. introduced a reference model-free monolithic odds ratio preference optimization algorithm. Some concurrent works attempt to introduce intrinsic knowledge constraints into preference alignment, either by on-the-fly fine-tuning LLMs to obtain relative qualities (Yu et al., 2024) or by defining a reward distance (Nvidia et al., 2024; Amini et al., 2024). These studies do not utilize tied preference to refine alignment, whereas our approach has the potential to be integrated with them as discussed in Section 7.

**Ternary preference models.** Ties frequently occur in ranked data, such as sports and examinations. For example, a soccer result can be classified simply as a home win, an away win, or a tie. The well-known BT model, which can be derived from the order statistics of the exponential distribution, cannot handle ties. Rao & Kupper (1967) corrected the BT model by assuming that small probability difference values would be declared ties. Kuk (1995) applied this approach to football. Davidson (1970) provided an ad hoc correction to the BT model for ties. Dewart & Gillard (2018) applied the BT model to cricket, where draws occur but do not depend strongly on team strengths. Baker & Scarf (2020) used discrete distributions, principally the geometric distribution, to obtain a modified BT model that allows for tie ranks. These studies lay a robust theoretical foundation for addressing ties in ranking results, facilitating the connection between LLM alignment and preference data.

## 3 PRELIMINARIES

### 3.1 THE BRADLEY-TERRY MODEL

The Bradley-Terry (BT) (Bradley & Terry, 1952) model is a probability model for the outcome of pairwise comparisons between instances, teams, or objects. It estimates the probability that the rank order $i \succ j$ is true, where the symbol $\succ$ represents a preference or rank relation, such as instance $i$ being preferred over $j$, or $i$ beating $j$, depending on the application.

The computation of BT model can be represented by Equation 1, where the positive strength owned by two competitors are denoted by $\lambda_1$ and $\lambda_2$ respectively, and $r_{12}$ represents the probability that the first competitor obtains a higher rank than the second one in comparison.

$$r_{12} = \frac{\lambda_1}{\lambda_1 + \lambda_2} \tag{1}$$

Define $d_{12} = \ln \lambda_1 - \ln \lambda_2$, which represents the strength difference of two competitors in logarithmic form. Following Rao & Kupper (1967), Equation 1 can be written into the form of Equation 2, where $\mathrm{sech}(\cdot)$ denotes the hyperbolic secant function. This relation shows that the preference probability $r_{12}$ depends only on $d_{12}$.

$$r_{12} = \frac{1}{4} \int_{-(\ln \lambda_1 - \ln \lambda_2)}^{\infty} \mathrm{sech}^2(y/2) \, dy = \frac{1}{4} \int_{-d_{12}}^{\infty} \mathrm{sech}^2(y/2) \, dy \tag{2}$$

### 3.2 HUMAN PREFERENCE MODELING

Following the BT model, the human preference distribution $p^*$ is formulated in Equation 3, where $r^*$ is some latent reward model, $r^*(x, y)$ denotes the reward for response $y$ given an input prompt

$x$, and $\sigma(\cdot)$ is the Sigmoid function. The variables $x$, $y_1$ and $y_2$ are drawn from a preference dataset $\mathcal{D} = \{x^i, y_1^i, y_2^i\}$, where $y_1$ is the preferred response and $y_2$ is the dispreferred one. The term $\exp(r^*(x, y))$ signifies the strength $\lambda$ of responses following Equation 1, acknowledging that the reward could be negative,

$$p^*(y_1 \succ y_2 | x) = \frac{\exp(r^*(x, y_1))}{\exp(r^*(x, y_1)) + \exp(r^*(x, y_2))} = \sigma(r^*(x, y_1) - r^*(x, y_2)). \tag{3}$$

Subsequently, a reward model that mirrors human preferences can be trained using the method of maximum likelihood estimation (Schulman et al.),

$$\max_\theta \mathbb{E}_{(x, y_1, y_2) \sim \mathcal{D}} \big[ \log \sigma \big( r_\theta(x, y_1) - r_\theta(x, y_2) \big) \big]. \tag{4}$$

### 3.3 DIRECT PREFERENCE OPTIMIZATION

In RLHF (Schulman et al.), the goal is to maximize the expectation of rewards under the KL divergence constraint,

$$\max_{\pi_\theta} \mathbb{E}_{x \sim D, y \sim \pi_\theta(y|x)}[r_\psi(x, y)] - \beta \log \frac{\pi_\theta(y|x)}{\pi_{\text{ref}}(y|x)}. \tag{5}$$

The optimal solution to this problem satisfies the following relationship (Rafailov et al.):

$$r(x, y) = \beta \log \frac{\pi^*(y|x)}{\pi_{\text{ref}}(y|x)} + \beta \log Z(x), \tag{6}$$

where $Z(x) = \Sigma_y \pi_{\text{ref}}(y|x) \exp\left(\frac{1}{\beta} r(x, y)\right)$ only depends on prompt $x$.

By integrating this relation into Equation 4, the loss function of DPO can be expressed as shown in Equation 7, which is incapable of addressing tied preference data.

$$\mathcal{L}_{\text{DPO}}(\pi_\theta; \pi_{\text{ref}}) = -\mathbb{E}_{(x, y_1, y_2) \sim \mathcal{D}} \left[ \log \sigma \left( \beta \log \frac{\pi_\theta(y_1|x)}{\pi_{\text{ref}}(y_1|x)} - \beta \log \frac{\pi_\theta(y_2|x)}{\pi_{\text{ref}}(y_2|x)} \right) \right]. \tag{7}$$

## 4 TIE-RANK ORIENTED DIRECT PREFERENCE OPTIMIZATION

To handle ties in preferences modeling, we introduce a buffer in the integral interval of Equation 2, which is inspired by Rao & Kupper (1967). We call the new preference model **Ti**e-rank **O**riented **BT** (TOBT) model. Based on this model, we propose a novel preference alignment algorithm, **Ti**e-rank **O**riented **D**irect Preference **O**ptimization (TODO).

### 4.1 TIE-RANK ORIENTED BT MODEL

As shown in Equation 2, $d_{12}$ represents the preference difference between two instances, and $d_{12} > 0$ means the first instance is preferred. To handle ties, we impose higher requirements on the comparison by introducing a positive number $\alpha$, and then the preferred relation is determined by $d_{12} > \alpha$. That is, the ranking probability $r_{12}$ becomes Equation 8,

$$r_{12} = \frac{1}{4} \int_{-(\ln \lambda_1 - \ln \lambda_2) + \alpha}^{\infty} \text{sech}^2(y/2) \, dy \tag{8}$$

Then, the probability that two instances share a tie relation is $1 - r_{12} - r_{21}$, which is denoted by $r_{(12)}$, and can be expressed in Equation 9,

$$r_{(12)} = \frac{1}{4} \int_{-(\ln \lambda_1 - \ln \lambda_2) - \alpha}^{-(\ln \lambda_1 - \ln \lambda_2) + \alpha} \text{sech}^2(y/2) \, dy \tag{9}$$

Intuitively, the parameter $\alpha$ encapsulates the inherent uncertainty and noise in the strength values $\lambda_1$ and $\lambda_2$, which are ubiquitous in LLM alignment. Preference datasets for alignment often rely on human labeling (Chiang et al., 2024), LLM-as-judge assessments (Cui et al., 2023), or reward model scoring (Nvidia et al., 2024), introducing label noise due to inconsistencies among human annotators or the inherent variability of LLMs in approximating human preferences. Accounting for this

uncertainty and noise in responses evaluation likely contributes to TODO's improved performance on binary preference datasets, as detailed in Section 6.

Following Equations 8 and 9, the TOBT model can be represented by Equations 10 and 11, where $\phi = \exp(\alpha)$ and $\phi > 1$. The detailed derivation process is provided in Appendix A.2.

$$r_{12} = \frac{\lambda_1}{\lambda_1 + \phi\lambda_2} \tag{10}$$

$$r_{(12)} = \frac{\lambda_1\lambda_2(\phi^2 - 1)}{(\lambda_1 + \phi\lambda_2)(\phi\lambda_1 + \lambda_2)} \tag{11}$$

### 4.2 OBJECTIVE FUNCTION OF TODO

Following Equations 10 and 11, the tie-rank oriented human preference distribution $p^*$ can be expressed by Equation 12 and Equation 13.

$$p^*(y_1 \succ y_2|x) = \frac{\exp(r^*(x, y_1))}{\exp(r^*(x, y_1)) + \phi\exp(r^*(x, y_2))} \tag{12}$$

$$p^*(y_1 \equiv y_2|x) = \frac{\exp(r^*(x, y_1))\exp(r^*(x, y_2))(\phi^2 - 1)}{\Big(\exp(r^*(x, y_1)) + \phi\exp(r^*(x, y_2))\Big)\Big(\exp(r^*(x, y_2)) + \phi\exp(r^*(x, y_1))\Big)} \tag{13}$$

Equation 12 represents the possibility of treating $y_1$ as the preferred response and $y_2$ as the dispreferred response in pairwise data based on the TOBT model. Following Equation 12, the objective $\mathcal{L}^p_{\text{TODO}}$ can be written as Equation 14, where $\mu = r_\theta(x, y_1) - r_\theta(x, y_2)$ represents the reward difference of two responses $y_1$ and $y_2$. Because the $Z(x)$ in implicit reward $r_\theta(x, y)$ only depends on $x$, the difference results of $\mu$ can be equivalently expressed as $\mu = \beta\log\frac{\pi_\theta(y_1|x)}{\pi_{\text{ref}}(y_1|x)} - \beta\log\frac{\pi_\theta(y_2|x)}{\pi_{\text{ref}}(y_2|x)}$. The superscript $p$ in $\mathcal{L}^p_{\text{TODO}}$ denotes that this formulation is the objective of TODO when the responses in a pair exhibit a clear preference, rather than being tied. The detailed derivation process is provided in Appendix A.4.

$$\begin{aligned}\mathcal{L}^p_{\text{TODO}}(\pi_\theta; \pi_{\text{ref}}) &= -\mathbb{E}_{(x,y_1,y_2)\sim\mathcal{D}}\big[\log\sigma(\mu - \alpha)\big] \\ &= -\mathbb{E}_{(x,y_1,y_2)\sim\mathcal{D}}\bigg[\log\sigma\Big(\beta\log\frac{\pi_\theta(y_1|x)}{\pi_{\text{ref}}(y_1|x)} - \beta\log\frac{\pi_\theta(y_2|x)}{\pi_{\text{ref}}(y_2|x)} - \alpha\Big)\bigg].\end{aligned} \tag{14}$$

Equation 13 represents the possibility of treating pairwise responses $y_1$ and $y_2$ as a tied pair based on the TOBT model. Following Equation 13, the objective $\mathcal{L}^t_{\text{TODO}}$ can be written as Equation 15. The superscript $t$ in $\mathcal{L}^t_{\text{TODO}}$ signifies that this represents TODO's objective when the pair is tied. The detailed derivation process is provided in Appendix A.5.

$$\mathcal{L}^t_{\text{TODO}}(\pi_\theta; \pi_{\text{ref}}) = -\mathbb{E}_{(x,y_1,y_2)\sim\mathcal{D}}\bigg[\log\frac{\exp(2\alpha) - 1}{(1 + \exp(\mu + \alpha))(1 + \exp(-\mu + \alpha))}\bigg] \tag{15}$$

Given a preference dataset $(x^i, y_1^i, y_2^i, \mathbb{I}^i) \in \mathcal{D}$, the indicator $\mathbb{I}^i$ is determined by Equation 16. Specifically, $\mathbb{I}^i = 0$ indicates a clear preference or quality difference between the two responses to the same prompt $x$, while $\mathbb{I}^i = 1$ represents that two responses $y_1^i$ and $y_2^i$ are tied. Then, the final loss $\mathcal{L}_{\text{TODO}}$ of TODO can be represented by Equation 17.

$$\mathbb{I}^i = \begin{cases} 1, & y_1^i \equiv y_2^i, \\ 0, & y_1^i \succ y_2^i. \end{cases} \tag{16}$$

$$\mathcal{L}_{\text{TODO}} = (1 - \mathbb{I})\mathcal{L}^p_{\text{TODO}} + \mathbb{I}\mathcal{L}^t_{\text{TODO}} \tag{17}$$

Compared to DPO, TODO introduces a margin $\alpha$ to shift the decision boundary towards $\alpha$ for paired data with a clear preference, while also accommodating ties. In contrast, methods based on the BT model struggle to handle tie data effectively.

### 4.3 THE EFFECT OF DIFFERENT PREFERENCE RELATIONS ON TODO UPDATES

To elucidate the dynamics of TODO parameter updates, we introduce a gradient-based analysis that distinguishes between scenarios where pairwise responses show a clear preference or are tied. The comprehensive derivation is detailed in Appendices A.6 and A.7.

**For the gradient update (18) for pairwise data with a clear preference**, the introduction of a positive small value $\alpha$ results in more substantial weight adjustments when the reward difference is misestimated compared to DPO. This refinement mitigates noise from narrow reward margins, allowing TODO to more effectively learn distinct preferences by concurrently enhancing the likelihood of the favored response $y_1$ and diminishing that of the unfavored response $y_2$.

$$\nabla_\theta \mathcal{L}^p_{\text{TODO}}(\pi_\theta; \pi_{\text{ref}}) = -\mathbb{E}_{(x,y_1,y_2)\sim\mathcal{D}}\left[ \underbrace{\beta\sigma(-\mu+\alpha)}_{\text{higher weight than DPO}} \left[\nabla_\theta \log(\pi(y_1|x)) - \nabla_\theta \log(\pi(y_2|x))\right]\right]. \quad (18)$$

**For the gradient update (19) for pairwise tie data**, $G(\mu) = \frac{\exp(-\mu+\alpha)-\exp(\mu+\alpha)}{(1+\exp(-\mu+\alpha))(1+\exp(\mu+\alpha))}$ is monotonically decreasing with respect to $\mu$, and $G(0) = 0$. When $\mu = 0$, two responses obtain the same rewards, and DPO continues to update policy models as per Equation 7, which shifts the distribution to reduce the likelihood of the "dispreferred" response $y_2$, potentially discarding valuable information. In contrast, TODO refrains from updating any parameters to maintain the consistent preference alignment of the tied responses.

When the estimated reward difference for tied responses is $\mu > 0$, suggesting $y_1$ has a higher reward than $y_2$, TODO's gradient update strategy will reduce the likelihood of $y_1$ and increase that of $y_2$. Conversely, if $\mu < 0$, the update will elevate the likelihood of $y_1$ and lower that of $y_2$, ensuring the preference consistency between the tied responses is preserved.

$$\nabla_\theta \mathcal{L}^t_{\text{TODO}}(\pi_\theta; \pi_{\text{ref}}) = -\mathbb{E}_{(x,y_1,y_2)\sim\mathcal{D}}\left[ \underbrace{G(\mu)\nabla_\theta \log(\pi(y_1|x))}_{\text{part 1}} + \underbrace{G(-\mu)\nabla_\theta \log(\pi(y_2|x))}_{\text{part 2}}\right]. \quad (19)$$

## 5 EXPERIMENTAL SETTINGS

### 5.1 MODELS AND DATASETS

**Models.** We select two different series of models, Mistral-7B (Jiang et al., 2023) and Llama 3-8B (AI@Meta, 2024), as our experimental backbone models. We select zephyr-sft-full[2] (Tunstall et al.) as the supervised fine-tuning (SFT) version of Mistral model and llama3-8b-sft-ultrachat[3] as the SFT version model of Llama 3 model. Both zephyr-sft-full and llama3-8b-sft-ultrachat are fine-tuned on Ultrachat-200k (Ding et al., 2023) dataset.

**Training datasets.** For the datasets used in the preference alignment process, we construct 20k-size datasets with different tie data proportions from Ultrafeedback (Cui et al., 2023). Responses sharing the same quality score are classified as tied. The quality score for each response is a weighted score across multiple assessment metrics, taking into account helpfulness, truthfulness, verbalized calibration and honesty. Each sampled dataset exhibits a tie data ratio that varies within the set {0, 0.1, 0.2, 0.3}. Other details of the training sets are shown in Appendix A.8.

**Evaluation benchmarks.** We first compare the effectiveness of preference modeling ability between DPO and TODO. To this end, we curate an in-distribution test set containing 1500 non-tied samples and select the Reward Bench (Lambert et al.) as an out-of-distribution dataset. Subsequently, we select a suite of well-established benchmark datasets to evaluate the models comprehensively: 1) MT Bench (Zheng et al., 2023), which contains open-ended questions designed to assess the multi-turn conversational capabilities and the ability to follow instructions of LLMs. 2) A diverse set of benchmarks containing Piqa (Bisk et al., 2019), ARC-c, ARC-e (Clark et al., 2018), Hellaswag (Zellers et al., 2019), MMLU (Hendrycks et al., 2021) and Winogrande (Sakaguchi et al., 2019), which collectively evaluate the language comprehension and reasoning faculties of LLMs.

---

[2] https://huggingface.co/alignment-handbook/zephyr-7b-sft-full
[3] https://huggingface.co/kykim0/llama3-8b-ultrachat-sft

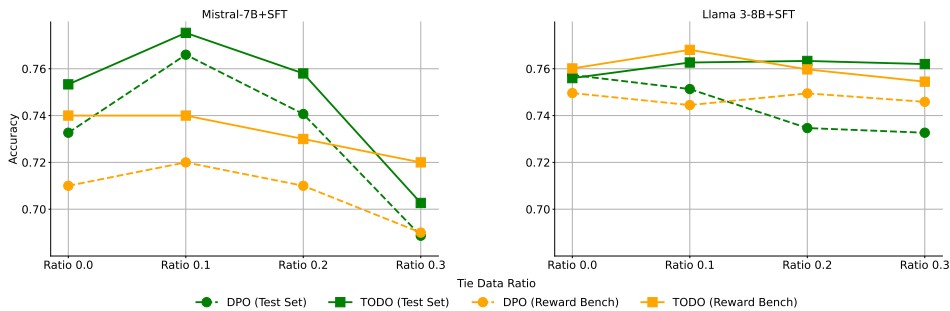

Figure 2: Accuracy of Mistral and Llama 3 models aligned with DPO and TODO on non-tie preference test set and Reward Bench. The X-axis denotes the proportion of tie data mixed in train set.

## 5.2 TRAINING SETTINGS

All comparative results are derived from training each model for 3 epochs on their respective training datasets. As demonstrated in the analysis presented in Appendix A.3, the model performance is relatively insensitive to variations in $\alpha$ within a reasonable range of $\alpha \in (0.1, 0.8)$, we set $\alpha = 0.5$ in TODO. Other hyperparameters are shown in Appendix A.9, where we adopt the settings from previous works (Saeidi et al.; Meng et al., 2024). We ensure the consistency of training hyperparameters among experiments for a fair comparison.

## 5.3 EVALUATION SETTINGS

**Accuracy of preference modeling.** In assessing the efficacy of models fine-tuned with DPO and TODO for preference modeling, we employ prediction accuracy as the primary evaluation metric. For each pair of data where the preference rank $y_1$ is favored over $y_2$, we calculate the predicted probabilities for each preference rank for pair data, adhering to Equations 12 and 13. A prediction is deemed accurate if the model assigns the highest probability to the scenario where $y_1$ is preferred over $y_2$ across all possible ranks.

**GPT based scoring in MT Bench.** For the evaluation in MT Bench, we use gpt-4-turbo-2024-04-09 (OpenAI, 2024b) to score generated results.

**Accuracy of other benchmarks.** For the evaluation of Piqa, ARC-c, ARC-e, Hellaswag, MMLU and Winogrande, we use Opencompass (Contributors, 2023) to assess final results, details of prompt templates and evaluation metrics can be found in Appendix A.10.

# 6 RESULTS AND ANALYSIS

## 6.1 TODO IMPROVES HUMAN PREFERENCE MODELING WITH TIE DATA

In this section, we assess the preference modeling capabilities of models trained with DPO and TODO across both in-distribution and out-of-distribution datasets, incorporating varying proportions of tie data in the training regimen. The in-distribution assessment is based on the above mentioned test set, while the out-of-distribution assessment utilizes the Reward Bench.

Figure 2 illustrates the accuracy results of Mistral and Llama 3 models aligned with DPO and TODO on the test set and Reward Bench, using training sets with varying proportions of tie data. Reward Bench contains pairwise responses divided into Chat, Chat hard, Safety, Reasoning and Prior preference data categories. We compute the average accuracy score of all categories to represent the final performance on Reward Bench. The detailed scores in each category of two series models on Reward Bench are provided in Appendix A.12.

We observe that both Mistral and Llama 3 models aligned with TODO generally achieve better performance than those aligned with DPO on both in-distribution and out-of-distribution data. Overall, when directly modeling human preferences mixing tie data, DPO often leads to sub-optimal results. TODO addresses this issue with more nuanced preference modeling. Experimental results demonstrate the effectiveness of the combinatorial optimization objectives in TODO.

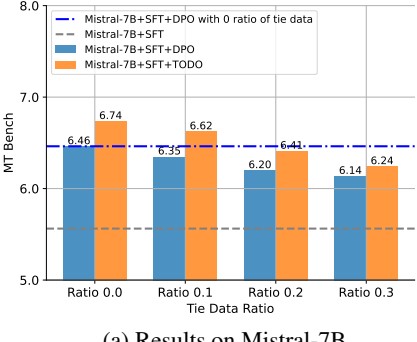 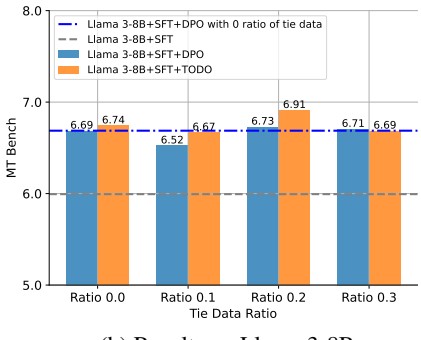

(a) Results on Mistral-7B        (b) Results on Llama 3-8B

Figure 3: MT Bench results of Mistral-7B and Llama 3-8B. The models are aligned with DPO and TODO using datasets with varying ratios of tie data.

## 6.2 TODO ENHANCES ALIGNMENT IN LLMS

In this section, we evaluate the effectiveness of DPO and TODO across different benchmarks to comprehensively demonstrate the alignment capabilities of LLMs.

As illustrated in Figures 3a and 3b, the scores on the MT Bench for models aligned with DPO and TODO reveal that TODO consistently outperforms DPO across all training sets for both the Mistral and Llama 3 models. Specifically, TODO achieves peak performance when using the binary preference dataset for the Mistral models and incorporating a 20% tie data ratio for the Llama 3 models, respectively.

Table 2 and Table 3 respectively show the results on Piqa, ARC-c, ARC-e, Hellaswag, MMLU and Winograrde benchmarks of the Mistral and Llama 3 models, where we highlight the best performance in underline, and mark the better performance between DPO and TODO with different ratio of tie data in **bold**. As demonstrated in these tables, TODO achieves better performance than DPO across all train sets on both two models, and obtain the best average performance when mixing 20% and 30% of tie data in Mistral and Llama 3 serials models, respectively.

**Optimal proportion of tie data enhances alignment.** Our experimental findings across various benchmarks underscore the critical role and efficacy of TODO in managing non-binary preference data. Notably, optimal performance on the MT Bench and six additional benchmarks is consistently achieved when a specific ratio of tie data is integrated with TODO. This observation indicates that the strategic incorporation of tie data, in conjunction with TODO, can markedly enhance the alignment capabilities of LLMs.

| Method | Tie Data Ratio | Piqa | ARC-c | ARC-e | MMLU | Hellaswag | Winogrande | Average |
|--------|----------------|-------|-------|-------|-------|-----------|------------|---------|
| SFT | ✗ | 80.09 | 51.33 | 74.97 | 60.26 | 75.69 | 73.64 | 69.33 |
| DPO | 0.0 | 77.15 | 49.87 | 69.56 | 59.44 | 73.55 | 67.88 | 66.24 |
| TODO | 0.0 | **81.01** | **54.59** | **78.44** | **60.16** | **79.01** | **72.93** | **71.02** |
| DPO | 0.1 | 79.87 | 53.82 | 77.21 | 60.07 | 79.04 | 73.48 | 70.58 |
| TODO | 0.1 | **80.25** | **54.68** | **77.76** | **60.46** | **79.27** | **73.95** | **71.06** |
| DPO | 0.2 | 80.25 | 55.02 | 77.51 | **59.79** | 79.08 | 73.01 | 70.78 |
| TODO | 0.2 | **81.07** | **55.11** | **78.52** | 59.40 | **79.14** | **73.80** | **71.17** |
| DPO | 0.3 | 75.73 | 48.15 | 68.84 | 59.41 | 71.97 | 69.30 | 65.57 |
| TODO | 0.3 | **77.37** | **51.59** | **71.75** | **59.79** | **74.36** | **69.38** | **67.37** |

Table 2: Results of Mistral-7B aligned with different methods trained with various ratios of tie data.

**TODO's superiority in binary preference alignment.** In addition to enhancing performance by incorporating tie data into the training regimen, TODO can also be effectively employed in scenarios involving purely binary preference alignment. The experimental outcomes, as depicted in Figure 3 for the MT Bench and Tables 2 and 3 for other benchmarks, consistently indicate that models fine-tuned with TODO outperform their counterparts optimized with DPO, even in the absence of any tie data.

| Method | Tie Data Ratio | Piqa | ARC-c | ARC-e | MMLU | Hellaswag | Winogrande | Average |
|--------|---------------|------|-------|-------|------|-----------|------------|---------|
| SFT | ✗ | 79.65 | 55.02 | 78.65 | 63.89 | 76.45 | 72.69 | 71.06 |
| DPO | 0.0 | 79.27 | 54.59 | 76.58 | 63.29 | 79.05 | 71.74 | 70.75 |
| TODO | 0.0 | **79.43** | **55.45** | **76.79** | **63.45** | 79.27 | **71.82** | **71.03** |
| DPO | 0.1 | 79.65 | 54.68 | 75.48 | 63.46 | **79.08** | **71.74** | 70.68 |
| TODO | 0.1 | 79.54 | **55.19** | **76.28** | **63.65** | 79.05 | 71.43 | **70.86** |
| DPO | 0.2 | 79.16 | 54.59 | 75.60 | **63.49** | 78.72 | 70.96 | 70.42 |
| TODO | 0.2 | **79.38** | 56.22 | **77.08** | 63.07 | **79.04** | **71.11** | **70.98** |
| DPO | 0.3 | 79.05 | 55.54 | 76.11 | 63.34 | 78.17 | 70.88 | 70.52 |
| TODO | 0.3 | 79.33 | 56.22 | **77.76** | **63.61** | **78.83** | **71.03** | 71.13 |

Table 3: Results of Llama 3-8B aligned with different methods with various ratios of tie data.

## 6.3 KEY FACTORS OF TODO IN HANDLING TIE DATA AND OBTAINING BETTER PERFORMANCE

In this section, we analyze the limitations of DPO and the advantages of TODO in handling tie data. We highlight two pivotal factors that TODO can enhance alignment: nuanced preference modeling and enriched diversity of information learned.

**TODO refines DPO by accurately handling tied data.** We consider the scenario where responses of equivalent quality are incorrectly labeled as binary preference data, as illustrated in Table 1. As stated in Section 4.2, $\mu$ represents the implicit reward margin between two responses during the training. For the $i$-th pair of responses, $y_1^i$ and $y_2^i$, which are essentially tied, DPO assigns a small value to $\mu^i$. Despite the trivial value of $\mu^i$, DPO unilaterally steers the policy model update by increasing the likelihood of one tied response $y_1^i$, while concurrently decreasing the likelihood of the other $y_2^i$. This biased update is a consequence of the binary BT model, following the gradient computation:

$$\nabla_\theta \mathcal{L}_{\text{DPO}} = -\mathbb{E}_{(x,y_1,y_2)\sim\mathcal{D}}\left[ \underbrace{\beta\sigma(-\mu)}_{\text{always positive}} \left[\nabla_\theta \log(\pi(y_1|x)) - \nabla_\theta \log(\pi(y_2|x))\right]\right]. \quad (20)$$

The gradient update direction as shown in Equation 20 is decided by the Sigmoid function and a positive $\beta$ value, which makes DPO always update policy model in one direction. Such a gradient computation, when applied to tied data, can result in erroneous updates and introduce noise into the binary preference modeling process, potentially culminating in a suboptimal policy. Both DPO and TODO assign a small reward margin for tied pair responses, as compared to utilizing data with clear preference distinction. However, TODO employs combinatorial optimization objectives for pairwise responses accounting for the uncertainty of preference difference, and effectively rectifying the issues DPO faces with tied data. This changes the update direction depending on the exact preference relation, ensuring preference learning consistency between two tied responses. Concurrently, the gradient computation for non-tied responses facilitates the learning of preferred or dispreferred relations. This distinction is crucial for effective preference modeling and can learn "prefer", "tie" and "disprefer" ranks. The exact reward margin changes of two models aligned with DPO and TODO using different ratios of tie data are provided in Appendix A.11 to support our analysis.

**Learning from diverse tied responses improves the performance.**

Both DPO and TODO are adept at ranking pairwise data with clear preference distinctions, distinguishing between "preferred" and "dispreferred" responses. However, when responses exhibit no clear preference or quality disparity, DPO's optimization strategy compels the policy model to skew towards one response at the expense of the other, resulting in unnecessary information loss for the unfavored response.

| Methods | Ultrafeedback (tie ratio 0) | Ultrafeedback (tie ratio 0.2) | Chatarena (tie ratio 0) | Chatarena (tie ratio 0.17) |
|---------|------|------|------|------|
| DPO | 76.40 | 75.00 | **77.47** | 76.33 |
| KTO | 73.80 | 74.27 | 74.80 | 74.80 |
| SimPO | **77.80** | 76.35 | 76.27 | 76.33 |
| ODPO | 76.78 | / | / | / |
| **TODO** | 77.20 | **76.40** | 76.80 | **78.47** |

Table 4: Test-set accuracy of Mistral-7B aligned with different methods and datasets.

TODO mitigates this by incorporating a tie rank and a novel optimization objective, enabling the policy model to evolve in accordance with the consistent preference trends of both responses. This mechanism allows both the content and format of the two responses to be learned simultaneously, enabling the model to capture more diverse information from the same amount of data.

## 6.4 COMPARISON AGAINST OTHER BINARY MODEL BASED APPROACHES.

To comprehensively assess the generalizability and efficacy of TODO, we conduct an exhaustive comparative analysis against several prominent binary alignment techniques, including DPO (Rafailov et al.), KTO (Ethayarajh et al., 2024), SimPO (Meng et al., 2024), and ODPO (Amini et al., 2024). Our evaluation transcends the limitations of a single dataset by leveraging both the Ultrafeedback dataset and the diverse, human-labeled Chatarena (lms, 2024) dataset, which encompasses multiple language pairs. Our experimental results on Mistral-7B, encompassing test set accuracies and MT Bench scores, are provided in Tables 4 and 5. In both datasets, TODO consistently outperforms other baselines when tie data is included, highlighting its advantages. Additionally, TODO with tie data surpasses methods without tie data in most cases, demonstrating the effectiveness of incorporating tie data. The experimental settings are provided in Appendix A.13.

## 7 DISCUSSION

**Potential integration of TOBT into other methods.** For RLHF-based methods, a straightforward integration way is to train a ternary reward model instead of the binary ones. The objective functions for the new reward model are composed of (12) and (13). To this end, a ternary-labeled preference dataset is needed, which is not difficult to obtain based on common annotation methods.

| Methods | Ultrafeedback (tie ratio 0) | Ultrafeedback (tie ratio 0.2) | Chatarena (tie ratio 0) | Chatarena (tie ratio 0.17) |
|---|---|---|---|---|
| DPO | 5.94 | 5.55 | 5.50 | 5.71 |
| KTO | 5.63 | 5.61 | 5.47 | 5.53 |
| SimPO | 5.56 | 5.69 | 5.21 | 5.28 |
| ODPO | **5.95** | / | / | / |
| **TODO** | 5.85 | **5.96** | **5.52** | **5.83** |

Table 5: MT Bench scores of Mistral-7B aligned with different methods and datasets.

For DPO-like methods, the integration varies by case. We take ODPO (Amini et al., 2024) as an example which adopts a loss function in the form of $-\log(\sigma(\mu - \Delta_r))$, where $\Delta_r$ introduces a preference margin depending on the ground-truth scores of the two inputs. By applying (15), we can obtain a new loss incorporating tie data $-\log\left(\frac{\exp(2\Delta_r)-1}{(1+\exp(\mu+\Delta_r))(1+\exp(-\mu+\Delta_r))}\right)$, the derivation of which should be similar to TODO.

**Extension for other tie-aware preference models.** The model's reliance on $r_1 - r_2$ rather than $r_1$ and $r_2$ individually is crucial, because the reward includes an untractable variable $Z(x)$ (Equation 6) which can be eliminated in $r_1 - r_2$. Some models (Glenn & David, 1960; Davidson, 1970) share this characteristic, offering potential for adaptation in LLM alignments. However, their effectiveness in this context has yet to be explored.

## 8 CONCLUSION AND FUTURE WORK

This study illuminates a limitation in LLM preference alignment: the inability of binary models like DPO to resolve ties in preference data. To overcome this, we integrate a tie ranking system into preference modeling, refine the BT model into the more robust TOBT model, and introduce the TODO algorithm. Our experimental results demonstrate that TODO consistently outperforms DPO across a range of evaluations. The success of TODO stems from its nuanced handling of ties and the robust TOBT model. This approach is not only limited to direct preference optimization but is also compatible with various online and offline preference optimization policies and can be employed for reward model training, which are left for future work.

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

# A  APPENDIX

## A.1  TIE DATA IN CURRENT PREFERENCE DATASETS

We conduct a statistical analysis of existing tie data across common preference data, as presented in Table 6. Current pairwise preference data are scored by LLMs or labeled by humans to establish preference rankings. Preference data scored by LLMs, commonly rely on the quality score considering multiple aspects to differentiate between preferred and dispreferred responses. Ultrafeedback (Cui et al., 2023), as a popular preference dataset, is scored based on GPT-4 feedback, and contains 383k pairwise responses. We find that 17.0% of the data pairs exhibit identical quality scores. The Ultrafeedback_binaried  (Tunstall et al.), a subset of this dataset, also includes 12.1% of tie data. This analysis underscores the prevalence of ties in preference datasets.

| Datasets | Tie data(%) |
| --- | --- |
| UltraFeedback | 17.0 |
| Ultrafeedback binarized | 12.1 |
| lmsys-chatbot_arena_conversations | 29.4 |

Table 6: Statistics on common preference datasets.

Unlike datasets featuring explicit quality scores, some preference datasets include only human-provided preference labels. Figure 4 illustrates two pairs of responses from the Chatbot Arena[4] (Chiang et al., 2024), judged by different users. Specifically, user793 prefers R1 and disfavors R2 for prompt 1. Conversely, user1245 exhibits no clear preference between the two responses for prompt 2, categorizing them as tied. The lmsys-chatbot_arena_conversations dataset, which collects the pairwise data from Chatbot Arena, reveals that 29.4% of the total samples are tied. This statistic highlights the significant presence of tie data in human-labeled preference data.

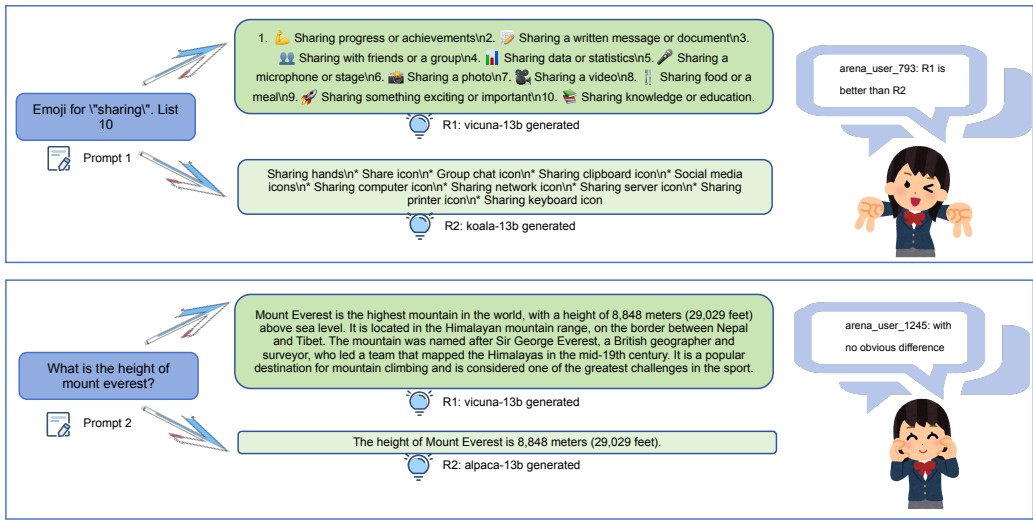

Figure 4: Human evaluation of pairwise responses across various prompts in the Chatbot Arena.

## A.2  THE TOBT MODEL

By substituting $t = \frac{y}{2}$ into $\frac{1}{4} \int \text{sech}^2(\frac{y}{2})dy$, we obtain that

$$F(t) = \frac{1}{2} \int \text{sech}^2(t)dt = \frac{1}{2} \tanh(t),$$  (21)

---

[4] https://lmsys.org/blog/2023-05-03-arena/

where $\tanh(t) = \frac{\exp(t)-\exp(-t)}{\exp(t)+\exp(-t)}$. Then $r_{12} = \frac{1}{4}\int_{-(\ln\lambda_1-\ln\lambda_2)+\alpha}^{\infty} \operatorname{sech}^2(y/2)\, dy$ can be written into $r_{12} = \frac{1}{2}\int_{t_1}^{\infty} \operatorname{sech}^2(t)\, dt$, where $t_1 = \frac{-(\ln\lambda_1-\ln\lambda_2)+\alpha}{2}$. We can obtain the representation of $r_{12}$ following Equation 22:

$$
\begin{aligned}
r_{12} &= \frac{\tanh(\infty) - \tanh(t_1)}{2} \\
&= \frac{1 - \tanh(t_1)}{2} \\
&= \frac{1 - \frac{\exp(t_1)-\exp(-t_1)}{\exp(t_1)+\exp(-t_1)}}{2} \\
&= \frac{1 - \frac{\exp(2t_1)+\exp(-2t_1)-2}{\exp(2t_1)-\exp(-2t_1)}}{2} \\
&= \frac{1 - \exp(-2t_1)}{\exp(2t_1) - \exp(-2t_1)} \\
&= \frac{1 - \exp(\ln\lambda_1 - \ln\lambda_2 - \alpha)}{\exp(\ln\lambda_2 - \ln\lambda_1 + \alpha) - \exp(\ln\lambda_1 - \ln\lambda_2 - \alpha)}.
\end{aligned}
\tag{22}
$$

By substituting $\alpha$ with $\ln\phi$ in the last line of Equation 22, the final presentation of $r_{12}$ can be represented by Equation 23:

$$
\begin{aligned}
r_{12} &= \frac{1 - \exp(\ln\lambda_1 - \ln\lambda_2 - \ln\phi)}{\exp(\ln\lambda_2 - \ln\lambda_1 + \ln\phi) - \exp(\ln\lambda_1 - \ln\lambda_2 - \ln\phi)} \\
&= \frac{1 - \frac{\lambda_1}{\phi\lambda_2}}{\frac{\phi\lambda_2}{\lambda_1} - \frac{\lambda_1}{\phi\lambda_2}} \\
&= \frac{\lambda_1}{\lambda_1 + \phi\lambda_2}.
\end{aligned}
\tag{23}
$$

For the $r_{(12)} = \frac{1}{4}\int_{-(\ln\lambda_1-\ln\lambda_2)-\alpha}^{-(\ln\lambda_1-\ln\lambda_2)+\alpha} \operatorname{sech}^2\left(\frac{y}{2}\right) dy$, we can rewrite it into $r_{(12)} = \frac{1}{2}\int_{t_2}^{t_1} \operatorname{sech}^2(t)\, dt$, in which $t_1 = \frac{-(\ln\lambda_1-\ln\lambda_2)+\alpha}{2}$ and $t_2 = \frac{-(\ln\lambda_1-\ln\lambda_2)-\alpha}{2}$. Then we compute $r_{(12)}$ according to Equation 24:

$$
\begin{aligned}
r_{(12)} &= \frac{1}{2}(\tanh(t_1) - \tanh(t_2)) \\
&= \frac{(\exp(t_1) - \exp(-t_1)(\exp(t_2) + \exp(-t_2))) - (\exp(t_2) - \exp(-t_2))(\exp(t_1) + \exp(-t_1))}{2(\exp(t_1) + \exp(-t_1))(\exp(t_2) + \exp(-t_2))} \\
&= \frac{\exp(t_1 - t_2) - \exp(t_2 - t_1)}{\exp(t_1 + t_2) + \exp(t_1 - t_2) + \exp(t_2 - t_1) + \exp(-t_1 - t_2)} \\
&= \frac{\exp(\alpha) - \exp(-\alpha)}{\exp(-(\ln\lambda_1 - \ln\lambda_2)) + \exp(\alpha) + \exp(\alpha) + \exp(\ln\lambda_1 - \ln\lambda_2)}.
\end{aligned}
\tag{24}
$$

By substituting $\alpha$ with $\ln\phi$ in the last line of Equation 24, we can obtain the final representation of $r_{(12)}$ as given in Equation 25:

$$
\begin{aligned}
r_{(12)} &= \frac{\phi - \frac{1}{\phi}}{\frac{\lambda_2}{\lambda_1} + \phi + \frac{1}{\phi} + \frac{\lambda_1}{\lambda_2}} \\
&= \frac{\lambda_1\lambda_2(\phi^2 - 1)}{\phi\lambda_2^2 + \phi^2\lambda_1\lambda_2 + \lambda_1\lambda_2 + \phi\lambda_1^2} \\
&= \frac{\lambda_1\lambda_2(\phi^2 - 1)}{(\lambda_1 + \phi\lambda_2)(\lambda_2 + \phi\lambda_1)}.
\end{aligned}
\tag{25}
$$

## A.3 SELECTION OF $\alpha$ IN TODO

In this section, we elaborate on the process of selecting the optimal value for $\alpha$ in TODO. Initially, we generate a series of random values to emulate the expression $\beta\log\frac{\pi_\theta(y_1|x)}{\pi_{\text{ref}}(y_1|x)} - \beta\log\frac{\pi_\theta(y_2|x)}{\pi_{\text{ref}}(y_2|x)}$,

which represents the reward difference between two responses in the initial stage of preference alignment. Subsequently, we calculate the corresponding pairwise non-tied and tied losses across a range of $\alpha$ values. Figures 5a and 5b present the visualization of the results for two distinct value spans, illustrating the impact of $\alpha$ on the losses.

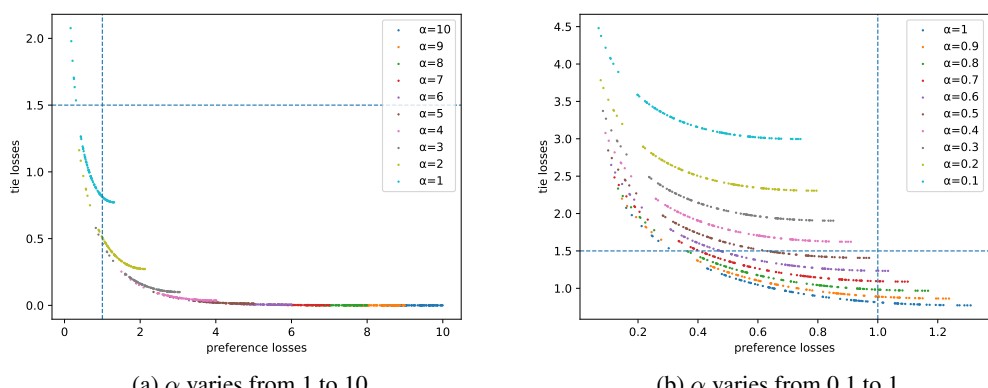

(a) $\alpha$ varies from 1 to 10          (b) $\alpha$ varies from 0.1 to 1

Figure 5: The initial preference and tie losses simulated with different $\alpha$ values.

A delicate balance between the non-tie loss and the tie loss is beneficial to the TODO performance. Our observations indicate that the non-tie and tie losses are interconnected through a power-law relation. As depicted in Figure 5, the preference loss sharply ascends with increasing values of $\alpha$, exceeding the concurrent growth in tie loss.

To achieve equilibrium between these two types of losses and to identify an appropriate value of $\alpha$ that maintains the preference loss near the levels observed in the original DPO (which implies that $\alpha$ should not be markedly distant from zero), we establish two thresholds: the initial tie loss in TODO is restricted to a maximum of 1.5, and the initial preference loss is restricted to a maximum of 1.0. These constraints are instrumental in guiding the selection of $\alpha$, ensuring that the optimization process effectively reconciles the competing objectives of preference and tie loss minimization.

Initially, we graph the variations in both preference and tie losses across a spectrum of $\alpha$ values, ranging from 1 to 10, as illustrated in Figure 5a. This visualization reveals that the preference loss using TODO is consistently higher than desired. Subsequently, we refine our approach by focusing on a narrower band of $\alpha$ values, specifically from 0.01 to 0.1. The detailed outcomes of this refined analysis are shown in Figure 5b.

Theoretically, the BT-model and our TOBT model both rely on the assumption that the reward difference $\mu = r(x, y_1) - r(x, y_2)$ follows the logistic distribution with unit variance. The mean of BT-model is 0 and the mean of TOBT is $\alpha > 0$. In the early training phase of LLM, $\mu$ is typically small in magnitude. Hence, a large $\alpha$ may produce a very large loss ($\mathbb{E} \log \sigma(\mu - \alpha)$) and leads to gradient vanishing, where $\sigma$ is the Sigmoid function. From this perspective, $\alpha < 1$ is beneficial since the gradient of $\sigma$ is less than 0.2 for $\alpha > 1$. On the other hand, $\alpha$ cannot be too close to 0 since the weights of tie data is propotional to $\exp(2\alpha)$. There is a clear tradeoff.

Empirically, we further compare the performance of TODO with different $\alpha$ on our preference testset and MT Bench besides in Tables 7 and 8, the results on MT Bench are scored by gpt-4o-2024-05-13 (OpenAI, 2024a). It shows that the performance is relatively not sensitive to the change of $\alpha$ within a reasonable range like $\alpha \in (0.1, 0.8)$, but degrade significantly when $\alpha$ is too large, which agrees with our theoretical analysis. This behaves like the tuning of learning rate, which can also lead to divergence beyond some threshold.

| Methods | Ultrafeedback (tie ratio 0) | Ultrafeedback (tie ratio 0.2) | Chatarena (tie ratio 0) | Chatarena (tie ratio 0.17) |
|---|---|---|---|---|
| TODO($\alpha = 0.1$) | 77.20 | 76.40 | 76.80 | **78.47** |
| TODO($\alpha = 0.2$) | **77.67** | 76.40 | 77.00 | 77.93 |
| TODO($\alpha = 0.5$) | 77.33 | **76.87** | **77.53** | 77.87 |
| TODO($\alpha = 0.8$) | Diverge | 75.47 | 77.27 | 76.40 |
| TODO($\alpha = 1.2$) | Diverge | 75.33 | Diverge | Diverge |

Table 7: Test-set accuracy in Mistral-7B model with different $\alpha$ values set in TODO.

| Methods | Ultrafeedback (tie ratio 0) | Ultrafeedback (tie ratio 0.2) | Chatarena (tie ratio 0) | Chatarena (tie ratio 0.17) |
|---|---|---|---|---|
| TODO($\alpha = 0.1$) | **5.85** | **5.96** | 5.52 | **5.83** |
| TODO($\alpha = 0.2$) | 5.44 | 5.94 | 5.53 | 5.58 |
| TODO($\alpha = 0.5$) | 5.73 | 5.91 | **5.69** | 5.78 |
| TODO($\alpha = 0.8$) | Diverge | 5.81 | 5.48 | 5.51 |
| TODO($\alpha = 1.2$) | Diverge | 5.41 | Diverge | Diverge |

Table 8: MT Bench scores in Mistral-7B model with different $\alpha$ values set in TODO.

## A.4 THE OBJECTIVE FUNCTION OF TODO WITH PAIRWISE DISTINGUISHED PREFERENCE DATA

It is straightforward to derive the TODO objective in pairwise data with distinct preference difference. Under the TOBT model as Equation 12, we can obtain the possibility of $y_1$ being preferred over $y_2$ following Equation 26 by substituting $\phi$ into $\exp(\alpha)$.

$$
\begin{aligned}
p^*(y_1 \succ y_2 | x) &= \frac{\exp(r^*(x, y_1))}{\exp(r^*(x, y_1)) + \phi \exp(r^*(x, y_2))} \\
&= \frac{\exp(r^*(x, y_1))}{\exp(r^*(x, y_1)) + \exp(\alpha) \exp(r^*(x, y_2))} \\
&= \frac{1}{1 + \exp(\alpha) \exp(r^*(x, y_2) - r^*(x, y_1))} \\
&= \frac{1}{1 + \exp(r^*(x, y_2) - r^*(x, y_1) + \alpha)} \\
&= \sigma(r^*(x, y_1) - r^*(x, y_2) - \alpha)
\end{aligned}
\tag{26}
$$

Recall that the (unavailable) ground-truth reward through is given as follows:

$$
r^*(x, y) = \beta \log \frac{\pi^*(y|x)}{\pi_{\text{ref}}(y|x)} + \beta \log Z(x).
\tag{27}
$$

Substituting Equation 27 into Equation 26, we derive the per-instance preference possibility as shown in Equation 28.

$$
p^*(y_1 \succ y_2 | x) = \sigma\left(\beta \log \frac{\pi^*(y_1|x)}{\pi_{\text{ref}}(y_1|x)} - \beta \log \frac{\pi^*(y_2|x)}{\pi_{\text{ref}}(y_2|x)} - \alpha\right)
\tag{28}
$$

## A.5 THE OBJECTIVE FUNCTION OF TODO WITH PAIRWISE TIE DATA

For instances of pairwise tie data, by utilizing the TOBT model as defined in Equation 13, we can compute the possibility that $y_1$ and $y_2$ are tied, as shown in Equation 29:

$$
\begin{aligned}
p^*(y_1 \equiv y_2 | x) &= \frac{\exp(r^*(x, y_1)) \exp(r^*(x, y_2))(\phi^2 - 1)}{(\exp(r^*(x, y_1)) + \phi \exp(r^*(x, y_2)))(\exp(r^*(x, y_2)) + \phi \exp(r^*(x, y_1)))} \\
&= \frac{\exp(r^*(x, y_1) + r^*(x, y_2))(\exp(2\alpha) - 1)}{(\exp(r^*(x, y_1)) + \exp(\alpha) \exp(r^*(x, y_2)))(\exp(r^*(x, y_2)) + \exp(\alpha) \exp(r^*(x, y_1)))}.
\end{aligned}
\tag{29}
$$

Which can be transferred into:

$$p^*(y_1 \equiv y_2|x) = \frac{\exp(2\alpha) - 1}{1 + \exp(r^*(x, y_1) - r^*(x, y_2) + \alpha) + \exp(r^*(x, y_2) - r^*(x, y_1) + \alpha) + \exp(2\alpha)},$$
(30)

by dividing $\exp(r^*(x, y_1) + r^*(x, y_2))$ in both the numerator and denominator. Because $\exp(2\alpha)$ can be expressed by $\exp(r^*(x, y_1) - r^*(x, y_2) + \alpha + r^*(x, y_2) - r^*(x, y_1) + \alpha)$, we can rewrite the denominator of the last line in Equation 30 into $(1 + \exp(r^*(x, y_1) - r^*(x, y_2) + \alpha))(1 + \exp(r^*(x, y_2) - r^*(x, y_1) + \alpha))$. Then we can obtain Equation 31.

$$p^*(y_1 \equiv y_2|x) = \frac{\exp(2\alpha) - 1}{(1 + \exp(r^*(x, y_1) - r^*(x, y_2) + \alpha))(1 + \exp(r^*(x, y_2) - r^*(x, y_1) + \alpha))}$$
(31)

Substituting Equation 27 into Equation 31, we obtain the per-instance tie possibility as shown in Equation 32, where $\mu = \beta \log \frac{\pi^*(y_1|x)}{\pi_{\text{ref}}(y_1|x)} - \beta \log \frac{\pi^*(y_2|x)}{\pi_{\text{ref}}(y_2|x)}$).

$$p^*(y_1 \equiv y_2|x) = \frac{\exp(2\alpha) - 1}{(1 + \exp(\mu + \alpha))(1 + \exp(-\mu + \alpha))}.$$
(32)

### A.6 THE GRADIENT OF TODO WITH PAIRWISE DISTINGUISHED PREFERENCE DATA

The gradient of TODO with pairwise distinguished preference data can be expressed by following equation:

$$\nabla_\theta \mathcal{L}^p_{\text{TODO}}(\pi_\theta; \pi_{\text{ref}}) = -\nabla_\theta \mathbb{E}_{(x, y_1, y_2) \sim \mathcal{D}}[\log \sigma(\mu - \alpha)],$$
(33)

where $\mu = \beta \log \frac{\pi_\theta(y_1|x)}{\pi_{\text{ref}}(y_1|x)} - \beta \log \frac{\pi_\theta(y_2|x)}{\pi_{\text{ref}}(y_2|x)}$. Then Equation 33 can be written into following form:

$$\nabla_\theta \mathcal{L}^p_{\text{TODO}}(\pi_\theta; \pi_{\text{ref}}) = -\mathbb{E}_{(x, y_1, y_2) \sim \mathcal{D}}[\frac{\sigma'(\mu - \alpha)}{\sigma(\mu - \alpha)} \nabla_\theta(\mu)].$$
(34)

Using the properties of Sigmoid function $\sigma'(x) = \sigma(x)(1 - \sigma(x))$ and $\sigma(-x) = 1 - \sigma(x)$, we obtain the final gradient, $\nabla_\theta \mathcal{L}^p_{\text{TODO}}(\pi_\theta; \pi_{\text{ref}}) = -\mathbb{E}_{(x, y_1, y_2) \sim \mathcal{D}}[\beta \sigma(\beta \log \frac{\pi_\theta(y_2|x)}{\pi_{\text{ref}}(y_2|x)} - \beta \log \frac{\pi_\theta(y_1|x)}{\pi_{\text{ref}}(y_1|x)} + \alpha)[\nabla_\theta \log(\pi(y_1|x)) - \nabla_\theta \log(\pi(y_2|x))]]$.

### A.7 THE GRADIENT OF TODO WITH PAIRWISE TIE DATA

The gradient of TODO with pairwise tie data can be expressed into following form:

$$\nabla_\theta \mathcal{L}^t_{\text{TODO}}(\pi_\theta; \pi_{\text{ref}}) = -\nabla_\theta \mathbb{E}_{(x, y_1, y_2) \sim \mathcal{D}}[\log(\frac{\exp(2\alpha) - 1}{f(\mu)})],$$
(35)

where $f(\mu) = (1 + \exp(-\mu + \alpha))(1 + \exp(\mu + \alpha))$, and $\mu = \beta \log \frac{\pi_\theta(y_1|x)}{\pi_{\text{ref}}(y_1|x)} - \beta \log \frac{\pi_\theta(y_2|x)}{\pi_{\text{ref}}(y_2|x)}$. Then Equation 35 can be rewritten into:

$$\nabla_\theta \mathcal{L}^t_{\text{TODO}}(\pi_\theta; \pi_{\text{ref}}) = -\mathbb{E}_{(x, y_1, y_2) \sim \mathcal{D}}[\frac{f'(\mu)}{f(\mu)} \nabla_\theta(\mu)].$$
(36)

We can derive the final gradient of TODO following Equation 37, where $G(\mu) = \frac{\exp(-\mu + \alpha) - \exp(\mu + \alpha)}{(1 + \exp(\mu + \alpha))(1 + \exp(-\mu + \alpha))}$.

$$\nabla_\theta \mathcal{L}^t_{\text{TODO}}(\pi_\theta; \pi_{\text{ref}}) = -\mathbb{E}_{(x, y_1, y_2) \sim \mathcal{D}}[G(\mu)[\nabla_\theta \log(\pi(y_1|x)) - \nabla_\theta \log(\pi(y_2|x))]]$$
(37)

For any value of $\mu$, we have $G(\mu) = -G(-\mu)$, indicating $G(\mu)$ is an odd function. Additionally, $G'(\mu)$, the first derivative of $G(\mu)$, is always negative derived from Equation 38, indicating $G(\mu)$ is monotonically decreasing with respect to $\mu$.

$$G'(\mu) = \frac{-(\exp(-\mu + \alpha) + \exp(\mu + \alpha))(1 + \exp(2\alpha) + 2\exp(-\mu + \alpha))}{(1 + \exp(\mu + \alpha))^2(1 + \exp(-\mu + \alpha))^2}.$$
(38)

## A.8 Construction of the training datasets

The original Ultrafeedback dataset contains about 64k prompts from diverse resources, including UltraChat(Ding et al., 2023), ShareGPT (Chiang et al., March 2023), Evol Instruct (Xu et al., 2023), TruthfulQA (Lin et al., 2022), FalseQA (Hu et al., 2023), and FLAN (Longpre et al., 2023). Each prompt is used to query multiple LLMs and generate 4 different responses, resulting in a total of 383k samples. Each response in pairwise data has its quality score provided by GPT-4 feedback. We sample 20k samples from these 383k samples to construct different training sets. To ensure consistency and fairness in data distribution, each sampling follows the original distribution of Ultrafeedback dataset, as shown in Table 9. We then construct datasets with different proportions of tie data, with the tie data ratios varying within the set $\{0, 0.1, 0.2, 0.3\}$.

| Data source | Evol instruct | False QA | FLAN | Sharegpt | TruthfulQA | Ultrachat |
|---|---|---|---|---|---|---|
| Percent(%) | 15.63 | 3.66 | 32.73 | 31.19 | 1.27 | 15.52 |

Table 9: Data source distribution in each sampled train set.

## A.9 Training hyperparameters

We use full-parameters fine-tuning when comparing different methods on Mistral-7B and Llama 3-8B models. We use Adam optimizer and the weight decay is set into 0. We use cosine learning rate scheduler, and the other detailed settings of DPO and TODO is shown in Table 10.

| Model | Learning rate | Batch size | $\beta$ |
|---|---|---|---|
| Mistral+SFT | 5e-7 | 64 | 0.01 |
| Llama 3+SFT | 1e-6 | 128 | 0.01 |

Table 10: Training hyperparameters settings of DPO and TODO.

## A.10 Evaluation metrics

For the evaluation based on OpenCompass, we use the default prompt template, and the specific metrics and evaluation mode settings are shown in Table 11. In this table, the PPL mode is used for multiple-choice tasks, utilizing the perplexity of each choice as the evaluation metric. The LL mode is used for the Winogrande task, where log likelihood is employed to evaluate task performance.

| Task | Piqa | ARC-c | ARC-e | MMLU | Hellaswag | Winogrande |
|---|---|---|---|---|---|---|
| Mode | PPL | PPL | PPL | PPL | PPL | LL |
| Metric | 0-shot | 0-shot | 0-shot | 5-shot | 0-shot | 0-shot |

Table 11: Evaluation details of multi downstream tasks, PPL represents accuracy based perplexity and LL represents accuracy based on log likelihood estimation.

## A.11 Reward margin changes during the training process

Figure 6 illustrates the reward margin changes of Mistral+SFT and Llama 3+SFT models aligned with DPO and TODO. We observe that the growth of the reward margin in DPO decelerates as the proportion of tie data within the training set increases for both the Mistral model as shown in Figure 6a, and the Llama 3 model as shown in Figure 6b. For models aligned with TODO, a similar deceleration in the growth of the reward margin is noted with an increasing proportion of tie data in the training dataset, as shown in Figure 6c for the Mistral model and in Figure 6d for the Llama 3 model. Furthermore, models aligned with TODO exhibit a more pronounced disparity in

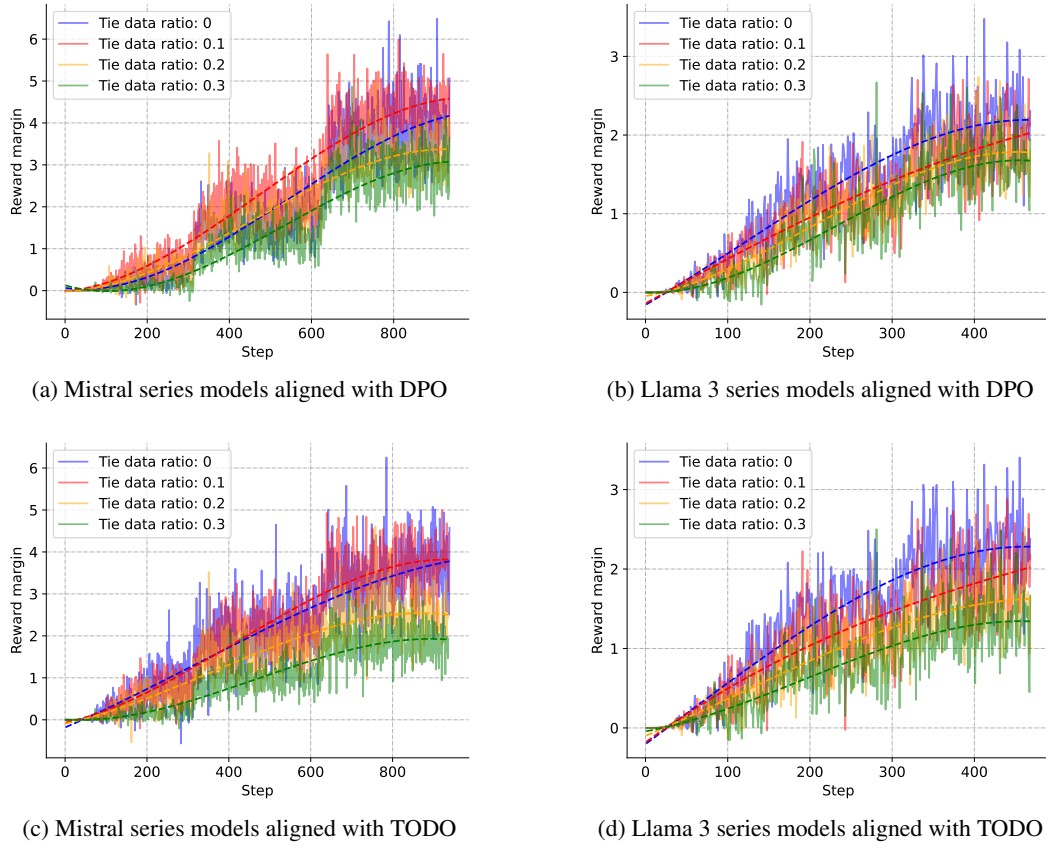

(a) Mistral series models aligned with DPO  (b) Llama 3 series models aligned with DPO

(c) Mistral series models aligned with TODO  (d) Llama 3 series models aligned with TODO

Figure 6: Reward margin changes of Mistral-7B and Llama 3-8B models aligned with DPO and TODO during the training process with varying tie data proportions. The dashed lines, sharing the same color as the solid lines, represent the fit of the reward margin.

reward margins when utilizing varying ratios of tie data, indicating a higher sensitivity of TODO to preference relations of the pairwise training data than DPO.

## A.12 DETAILED SCORES ON REWARD BENCH

In this section, we show the detailed scores of each category on Reward Bench. For the results, we highlight the best performance over all results in underline, and mark the performance better aligning with different ratio of tie data in **bold**. Table 12 and Table 13 show the performance of the Mistral+SFT and Llama 3+SFT models aligned with DPO and TODO on Reward Bench.

## A.13 MORE EXPERIMENTAL RESULTS AGAINST OTHER BINARY MODEL BASED METHODS

We compare the proposed TODO method with KTO (Ethayarajh et al., 2024), SimPO (Meng et al., 2024), and ODPO (Amini et al., 2024) on Mistral-7B evaluated on MT Bench and our preference testsets. Besides, we construct another Chatarena dataset[5], sourced from diverse human-labeled open data (lms, 2024). Chatarena captures the diversity and complexity of real-world human preferences across 96 different languages. It includes pairs with clear preference differences as well as ties. In our experiments, we used a training set of 20,000 pairs with tie data ratios of 0 and 0.17 (the natural tie ratio of this dataset) and a test set of 1,500 randomly selected samples.

---

[5]https://huggingface.co/datasets/irisxx/chatarena_tied

| Method | Tie Data Ratio | Chat | ChatHard | Safety | Reasoning | Prior | Average |
|---|---|---|---|---|---|---|---|
| SFT | ✗ | 66.48 | 54.61 | 35.73 | 62.94 | 40.20 | 51.99 |
| DPO | 0.0 | 91.34 | **66.45** | 75.79 | **72.36** | 49.23 | 71.03 |
| TODO | 0.0 | **93.85** | 64.25 | **76.88** | 72.17 | **61.07** | **73.64** |
| DPO | 0.1 | 91.9 | **67.98** | **73.64** | 72.18 | 53.67 | 71.87 |
| TODO | 0.1 | **94.13** | 64.69 | 71.80 | **73.43** | **64.25** | **73.66** |
| DPO | 0.2 | 91.34 | 64.47 | 78.27 | **74.37** | 49.83 | 71.66 |
| TODO | 0.2 | **95.81** | **64.69** | **78.49** | 62.30 | **61.63** | **72.58** |
| DPO | 0.3 | 85.75 | 64.04 | 75.78 | **75.01** | 45.56 | 69.23 |
| TODO | 0.3 | **89.94** | **65.13** | **77.13** | 73.37 | **57.06** | **72.53** |

Table 12: Mistral model results on Reward Bench trained with different ratios of tie data.

| Method | Tie Data Ratio | Chat | ChatHard | Safety | Reasoning | Prior | Average |
|---|---|---|---|---|---|---|---|
| SFT | ✗ | 67.60 | 55.48 | 41.89 | 65.17 | 43.26 | 54.68 |
| DPO | 0.0 | 93.3 | **67.32** | 75.18 | **84.23** | 54.78 | 74.96 |
| TODO | 0.0 | **93.58** | 65.79 | **77.66** | 83.32 | **59.72** | **76.01** |
| DPO | 0.1 | 93.85 | 63.82 | 77.14 | 83.52 | 53.91 | 74.45 |
| TODO | 0.1 | **96.37** | **64.69** | **79.32** | **84.08** | **59.59** | **76.81** |
| DPO | 0.2 | 91.62 | 66.45 | **78.18** | 84.80 | 53.70 | 74.95 |
| TODO | 0.2 | **92.74** | **66.89** | 75.50 | 84.37 | **60.37** | **75.97** |
| DPO | 0.3 | 89.39 | **65.35** | 78.31 | **87.02** | 52.87 | 74.59 |
| TODO | 0.3 | **93.02** | 64.91 | 74.35 | 84.64 | **60.33** | **75.45** |

Table 13: Llama 3 model results on Reward Bench trained with different ratios of tie data.

The results on MT Bench are scored by gpt-4o-2024-05-13 (OpenAI, 2024a). For ODPO, distinct quality scores are required for the chosen and rejected samples in a pair, a condition only met in the Ultrafeedback dataset without tie data. Therefore, we only report ODPO results under the tie data ratio 0 setting in Ultrafeedback. For each method's specific hyperparameter settings, we follow the configurations used in previous work (Saeidi et al.; Meng et al., 2024; Amini et al., 2024; Ethayarajh et al., 2024). As shown in Table 4 and Table 5, TODO achieves the best performance compared with other methods in the presence of tie data. If only binary preference data is available, TODO still delivers competitive performance.

