# OpenReview forum: "TODO: Enhancing LLM Alignment with Ternary Preferences"
_ICLR.cc/2025/Conference — ICLR 2025 Poster_

### Official Review · Reviewer_CZTq · 2024-10-28

**Soundness:** 3
**Presentation:** 3
**Contribution:** 3
**Rating:** 6
**Confidence:** 4

**Summary:**

Existing LLM alignment algorithms rely on the binary BradleyTerry model, which is hard to capture ties in preference. To address this issue, this paper proposes Tie-rank Oriented Direct Preference Optimization (TODO), an alignment algorithm that  leverages TOBT’s ternary ranking system.

TODO rewrites the ranking probability in BradleyTerry to add tie instances and derive the loss function of TODO. Experimental results verify the effectiveness of TODO.

**Strengths:**

The strengths of this paper can be listed as follows:
1. The motivation of this paper is clear and reasonable. Starting with the improvement of the BT model, this paper introduces the tie instance. This research idea is consistent with theoretical improvements of BT model.
2. The paper writing is good. I think it is easy to follow the authors' research ideas.
3. Experimental settings are detailed and well organized.

**Weaknesses:**

The weaknesses of this paper can be listed as follows:
1. **Limited baselines.** This paper only provides the comparison of experimental results between TODO and DPO, which is relatively limited. I think it is necessary to supplement some other algorithms as baselines, e.g. SimPO (https://arxiv.org/abs/2405.14734).
2. **Limited improvements.** The improvement of TODO over DPO is limited as shown in table 2 and table 3.

**Questions:**

I am also very curious that if the performance of LLM alignment will improve if you just simply filter out the tie preference pairs out of the training set.

---

> ### Author Response · Authors · 2024-11-24
> **Response to Reviewer CZTq[1/1]**
>
> We appreciate your detailed review and valuable feedback, we hope the following detailed responses could address your concerns.
>
> **W1: This paper only provides the comparison of experimental results between TODO and DPO, which is relatively limited. I think it is necessary to supplement some other algorithms as baselines, e.g. SimPO**
>
> - We present experimental results in **Tables R1 and R2** in our **general response** for the comparison of TODO with some recent approaches, including KTO, ODPO, and SimPO. In scenarios involving tie data, TODO consistently delivers superior performance in most cases. Notably, SimPO only surpasses TODO in test set accuracy when aligned with Ultrafeedback (tie ratio 0 settings), as illustrated in Table R1. In all other settings, TODO consistently outperforms SimPO.
>
> -  The discussion of TODO against other methods on both two datasets has been added in Section 7, the supporting experimental results have been added in Appendix A.13.
>
> **W2: The improvement of TODO over DPO is limited as shown in table 2 and table 3.**
> - Tables 2 and 3 in the manuscript present results for reasoning-heavy or knowledge-intensive tasks (e.g. ARC-c, HellaSwag, MMLU), which are not primarily designed for alignment evaluation. Previous research has shown that common alignment methods can often reduce LLM performance on these tasks, a phenomenon known as the "alignment tax" [1-3]. Through these tables, we illustrate that TODO incurs a smaller alignment tax compared to DPO.
>
> - For benchmarks specifically aimed at human preference alignment, such as MT Bench and Reward Bench, our method demonstrates notable improvements over DPO. Please see **Figure 2 and Figure 3** in the manuscript and **Tables R1 and R2** in our general response for details.
>
>
>
> **Q1: I am also very curious that if the performance of LLM alignment will improve if you just simply filter out the tie preference pairs out of the training set**
> - We conduct experiments by filtering out tie data, corresponding to the "tie ratio 0" in **Tables R1 and R2** in our **general response**. The results demonstrate that maintaining an appropriate ratio of tie data enhances TODO's performance.
>
>
> #### **References:**
>
> [1] Meng, Yu, Mengzhou Xia, and Danqi Chen. “SimPO: Simple Preference Optimization with a Reference-Free Reward.”
>
> [2] Ouyang, Long, Jeff Wu, Xu Jiang, Diogo Almeida, Carroll L. Wainwright, Pamela Mishkin, Chong Zhang, et al. “Training Language Models to Follow Instructions with Human Feedback.”
>
> [3] Lin, Yong, Hangyu Lin, Wei Xiong, Shizhe Diao, Jianmeng Liu, Jipeng Zhang, Rui Pan, et al. “Mitigating the Alignment Tax of RLHF.”

---

> ### Author Response · Authors · 2024-11-27
> **A Kind Reminder**
>
> Dear Reviewer  **CZTq**,
>
> We extend our heartfelt thanks for the insightful feedback you have provided, which has significantly enhanced our manuscript. As we approach the conclusion of the discussion phase, please feel free to share any additional concerns and we would be more than happy to address them.
>
> Kind regards,
>
> The Authors

---

> > ### Comment · Reviewer_CZTq · 2024-11-27
> > **Official Response to Authors**
> >
> > I really appreciate the complementary experiments and the efforts made by the authors. The reponse by authors helps me correct some misunderstandings. However, I still have some concerns about the paper.
> >
> > 1. Compared with algorithms such as DPO, I think TODO requires additional annotation resources. Ultrafeedback provides pointwise evaluations for different responses, which can help classify ties. However, for some other datasets, if we can only get pairwise comparisons of responses, we will need extra efforts to classify ties in preference.
> >
> > 2. From my practical experience with DPO, I believe that the training of DPO itself is relatively unstable, and the results are easily disturbed by the training data and hyperparameter settings. Therefore, although additional experiments have shown that TODO can outperform other baselines on MT Bench in most cases, I do not think the improvement provided in Table R1 and R2 is sufficient. Perhaps on some other datasets or alignment tasks, TODO can achieve a more significant performance improvement.
> >
> > In conclusion, I maintain my evaluation and scores now.

---

> ### Author Response · Authors · 2024-11-29
> **Further clarification**
>
> Dear Reviewer CZTq,
>
> We are immensely grateful for the time and support you have provided. We have carefully considered your feedback and would like to offer further clarification.
>
> #### **1. Tie data is prevalent and requires no additional annotation resources**
>
> We would like to re-clarify that tie data is not derived from additional annotations but is a natural occurrence during the formation of preference data. For instance, another dataset Chatarena, as demonstrated in our general response, is sourced from human-labeled preference data, naturally exhibits a frequency of 17% for tie pairs. To accommodate DPO and its variants [1-5], these tie pairs are typically filtered out. However, what TODO does is leverage the diversity of responses from these discarded high-quality tie pairs to enhance alignment. This is the key motivation of our work, as previously articulated in our manuscript.
>
> #### **2. TODO is still effective with binary preference data**
>
> In our manuscript, we delved into the "tie ratio 0" setting, which exclusively employs binary data. Within this context, we have presented the comparison of DPO and TODO. Figures 2 and 3 highlight TODO's superiority over DPO in preference modeling and MT Bench scores, respectively. In terms of reasoning tasks, Table 2 shows that TODO's average accuracy (71.02) exceeds DPO's (66.24) and Table 3 reveals that TODO (71.03) continues to outperform DPO (70.75). Reviewer **BkLA** has acknowledged that *"TODO works better than DPO even without ties in the data."*
>
> #### **3. TODO is more stable than DPO w.r.t. hyperparamters and datasets**
>
> Prior to our main experiments, we also undertook preliminary tests with other hyperparameters. The following Table R10 displays the MT Bench results of Llama3 8-B for two distinct hyperparameter settings. Notably, TODO demonstrates greater stability than DPO with the same training data and varied training settings.
>
> **Table R10: MT Bench scores on Llama 3-8B with different hyperparameters**
>
> | **Align** | **Hyperparametrs**                              | **Ultrafeedback (tie ratio 0)** | **Ultrafeedback (tie ratio 0.2)** |
> | :---- | :------------------------------------------ | :-------------- | :---------------- |
> | DPO   | Batch size 64, lr 5e-6                      | 6.74            | 6.05              |
> | DPO   | Batch size 128, lr 1e-6 (in the manuscript) | 6.69            | 6.73              |
> | TODO  | Batch size 64, lr 5e-6                      | 6.79            | 6.73              |
> | TODO  | Batch size 128, lr 1e-6 (in the manuscript) | 6.71            | 6.91              |
>
> Taking into account various datasets, we have summarized the performance of both DPO and TODO on the Ultrafeedback and Chatarena datasets within our general response. As demonstrated in Table R11, TODO consistently outperforms DPO across different datasets, regardless of whether tie data is present or not, showcasing not only enhanced stability but also improved alignment of TODO.
>
> **Table R11: Results on MT Bench across different datasets on Mistral-7B model**
>
> | **Methods**  | **Ultrafeedback (tie ratio 0)** | **Ultrafeedback (tie ratio 0.2)** | **Chatarena (tie ratio 0)** | **Chatarena (tie ratio 0.17)** |
> | :------- | :-------------------------- | :-------------------------- | :---------------------- | :------------------------- |
> | DPO      | 5.84                        | 5.55                        | 5.50                    | 5.71                       |
> | TODO | **5.85**                        | **5.96**                    | **5.52**                | **5.83**                   |
>
> Thank you once again for your valuable feedback. We hope that this clarification could fully address your concerns. We believe that our work presents a significant and well-substantiated contribution, and we kindly ask you to reconsider your score. Your updated evaluation would mean a great deal to us.
>
> Kind regards,
>
> The Authors
>
> #### References
>
> [1] Amini, Afra, Tim Vieira, and Ryan Cotterell. “Direct Preference Optimization with an Offset.”
>
> [2] Ethayarajh, Kawin, Winnie Xu, Niklas Muennighoff, Dan Jurafsky, and Douwe Kiela. “KTO: Model Alignment as Prospect Theoretic Optimization.”
>
> [3] Nvidia, Bo Adler, Niket Agarwal, Ashwath Aithal, Dong H. Anh, Pallab Bhattacharya, Annika Brundyn, et al. “Nemotron-4 340B Technical Report.”
>
> [4] Pal, Arka, Deep Karkhanis, Samuel Dooley, Manley Roberts, Siddartha Naidu, and Colin White. “Smaug: Fixing Failure Modes of Preference Optimisation with DPO-Positive.”
>
> [5] Xu, Haoran, Amr Sharaf, Yunmo Chen, Weiting Tan, Lingfeng Shen, Benjamin Van Durme, Kenton Murray, and Young Jin Kim. “Contrastive Preference Optimization: Pushing the Boundaries of LLM Performance in Machine Translation.”

---

> > ### Comment · Reviewer_CZTq · 2024-11-30
> > **Official Response to Authors**
> >
> > Thank you so much for the additional results and explanations. I have no further questions.

---

> > > ### Author Response · Authors · 2024-12-03
> > > **Thanks for your support**
> > >
> > > Dear Reviewer CZTq,
> > >
> > > Thank you once again for your valuable feedback. We hope that our rebuttal has fully addressed your concerns. In our response and revised submission, we have:
> > >
> > > - Provided comprehensive comparison between TODO and DPO across various benchmarks utilizing both binary and ternary preference data. The results, which showcase higher accuracy in preference modeling, improved MT Bench scores, and reduced alignment tax in reasoning-related tasks, consistently demonstrate the advantages of TODO within the alignment domain.
> > > - Developed an additional real-world human preference dataset and evaluated our TODO against DPO, KTO, SimPO and ODPO. Results shown in general response underscores the efficacy of integrating tie data and the robust, consistent performance of our approach TODO across varied datasets.
> > > - Re-clarified how TODO leverages the diversity of responses from these discarded high-quality tie pairs to enhance alignment without additional annotation resources.
> > >
> > > We believe that our work delivers a substantial and significant contribution to the field. With the discussion period deadline drawing near, we would greatly appreciate it if you could reconsider the score. Your updated evaluation would mean a great deal to us.
> > >
> > > Best regards,
> > >
> > > The Authors

---

### Official Review · Reviewer_BkLA · 2024-10-31

**Soundness:** 3
**Presentation:** 3
**Contribution:** 3
**Rating:** 8
**Confidence:** 3

**Summary:**

This paper proposes the inclusion of ties in the DPO framework to be able to model preferences in a more nuanced way. It derives a ternary model for preferences analogously to the binary Bradley-Terry model, and defines a novel preference learning objective (TODO) accordingly. Experiments are conducted by preference-tuning Llama3 and Mistral with preferences from Ultrafeedback with varying proportions of ties, and evaluating them in and out of domain, and on downstream tasks. The new TODO objective outperforms DPO in preference modeling and downstream tasks across the bench.

**Strengths:**

1. The idea is well motivated, as ties are relevant especially in human preference elicitation and can be expected to become even more prevalent the stronger generative models get.
2. The experiments are strongly supporting the new objective, the results are convincing.
3. The proposed solution is based on a solid intuition and supported by detailed derivation.

**Weaknesses:**

The only claim that is empirically not supported is that TODO “exhibits better robustness against potential noise in binary preference data.” There is no experiment studying noise in preferences (ties do not need to be caused by noise, they also present adequate equalness). One could easily set up an experiment with artificial noise added to preferences to test this hypothesis.

**Questions:**

1. What is the intuition for why it works better than DPO even without ties in the data?
2. What is the naturally occurring frequency of ties in the UltraFeedback data?
3. How does it compare empirically against related approaches, especially ODPO?

---

> ### Author Response · Authors · 2024-11-24
> **Response to Reviewer BkLA[1/1]**
>
> Thank you for your kind and constructive feedback. We would like to address each of your points individually.
>
>  **W1: The only claim that is empirically not supported is that TODO “exhibits better robustness against potential noise in binary preference data.” There is no experiment studying noise in preferences (ties do not need to be caused by noise, they also present adequate equalness). One could easily set up an experiment with artificial noise added to preferences to test this hypothesis.**
>
> - Following your suggestion, we created a binary preference dataset with noise by randomly selecting a sample from tied pairs in the Ultrafeedback dataset. We then train Mistral-7B on this dataset using DPO and TODO methods. The results, shown in **Table R9,** indicate that TODO performs better in this context. Kindly note that TODO achieves optimal performance when tie information is explicitly included.
>
>      **Table R9 : MT Bench scores with and without noise settings in DPO and TODO**
>
>     | DPO on Ultrafeedback with noise |TODO on Ultrafeedback with noise |TODO on Ultrafeedback with tie ratio 0.2 |
>     | ---------------- | ------------------------------ | -------------------------------- |
>     |5.55                           | 5.64                           | 5.96|
>
>
> **Q1: What is the intuition for why it works better than DPO even without ties in the data?**
>
> - Our method improves upon DPO by introducing a margin $\alpha$ in the decision boundary, even when tie data is absent. This adjustment changes the chosen probability to $\sigma(\mu-\alpha)$ instead of $\sigma(\mu)$. While omitting the margin might fit the training data better, it can lead to an overly complex decision boundary and potential overfitting. This concept is similar to the margin used in Support Vector Machines (SVMs), which has shown promising results. Additionally, as mentioned, ODPO applies varying margins for different inputs, also demonstrating effectiveness, which we will elaborate on in our response to **Q3**.
>
>
> **Q2: What is the naturally occurring frequency of ties in the UltraFeedback data?**
> - We have provided the statistics in Appendix  A.1 regarding the frequency of ties in UltraFeedback. Out of a total of 383k instances, the natural frequency of ties is **17.0%**.
>
>
> **Q3: How does it compare empirically against related approaches, especially ODPO?**
> - We present experimental results in **Tables R1 and R2** in our **general response** for the comparison of TODO with related approaches, including KTO, SimPO, and ODPO. In scenarios with tie data, TODO consistently outperforms other methods across both datasets. Without tie data, ODPO surpasses TODO in MT Bench scores, likely due to its dynamic margin compared to the static margin used in TODO. However, as noted in our **general response**, ODPO struggles with tie data because of its requirements on distinct quality scores. In contrast, TODO effectively leverages tie data to achieve better results. It would be an intriguing direction to integrate the dynamic margin concept from ODPO with the ternary preference model in future works.
>
> -  The discussion of TODO against other methods on both two dataset has been added in Section 7, the supporting experimental results have been added in Appendix A.13.

---

> > ### Comment · Reviewer_BkLA · 2024-11-26
> >
> > Thank you so much for the additional results and explanations! Both very convincing.

---

> > > ### Author Response · Authors · 2024-12-04
> > > **Thanks for your supportive feedback**
> > >
> > > We are deeply grateful for the time and dedication during discussion period. Your valuable comments has significantly enhanced our work. Thank you once again for your positive and supportive feedback.

---

### Official Review · Reviewer_iuqB · 2024-11-04

**Soundness:** 3
**Presentation:** 3
**Contribution:** 3
**Rating:** 5
**Confidence:** 3

**Summary:**

This work aims to directly model the fact that binary preference models such as the standard and well-known Bradley-Terry (BT) model cannot integrate ties and uncertainties when working with preference data. The main idea is thus the 'ternarization' of the original binary BT model that takes into account possible ties, where 'ties' are modeled via an additional parameter alpha. A large part of the paper is then dedicated to motivating and deriving the ternary (i.e., tie-aware) BT model, and then, based on that model on deriving and explaining TODO, a tie-aware extension of the original DPO algorithm for preference optimization. Experiments with two standard models (Mistral and LLaMA) and with a spectrum of standard datasets demonstrate the benefits of the TODO approach over its base binary DPO approach, even in setups without any ties. The authors then aim to provide some theoretical and empirically grounded interpretations on what makes TODO a more expressive and better-performing alternative to DPO.

**Strengths:**

- The paper is well written, and it is very easy to understand its core motivations and follow how the entire derivation of the tie-aware BT model and the corresponding TODO model unfolds from the main design assumptions.
- The provided results offer reasonable evidence on the benefits of TODO over DPO. The chosen evaluation data and models are adequate.
- The paper raises awareness on modeling ties in preference optimizations, where such valuable data often gets discarded under inadequate models or models with too strong assumptions (such as the binary BT) - taking into account finer granularity of preference data might yield further improvements in this line of work, and this paper sheds some new light on this link.

**Weaknesses:**

- One of the core issues with the work lies in the fact that other researchers in 'less recent times' have already attempted and derived generalised versions of the BT model, which better align with multi-class (i.e., non-binary) problems. Even a cursory search online reveals some very relevant literature which is not cited nor discussed in this work. I would require the authors to provide a thorough overview of that relevant work and discuss why a new (tie-aware) derivation might be needed here, and whether previous solutions could be repurposed for the extension of the original, binary BT model. For instance, I refer to the following relevant work here:
-- https://dl.acm.org/doi/pdf/10.5555/1248547.1248551
-- https://proceedings.neurips.cc/paper_files/paper/2004/hash/825f9cd5f0390bc77c1fed3c94885c87-Abstract.html
-- https://link.springer.com/article/10.1007/s10994-011-5240-0
See also Q1 related to this comment.

- Another concern is the lack of generality of TODO as it is derived directly as an extension of one concrete model: DPO. Following that, only DPO is also used as the only relevant baseline, and I see at least two potential issues here:
(i) While the authors state at the very end of the paper that the approach is "(...) not only limited to direct preference optimization but is also compatible with various online and offline preference optimization policies and can be employed for reward model training...", already this work should have coupled the idea with another preference optimization method to show this decoupling. In essence, the authors should both theoretically and empirically discern between making 'ternary decisions' and deriving enhanced preference optimization algorithms that incorporate the possibility to do ternary decisions.
(ii) DPO is used as the only baseline, while there is a plethora of other online and offline preference optimization algorithms out there, where some of them should have been included as baselines and reference points. For instance, the KTO approach disposes of the need to even collect preference data and model preference decisions as part of the process (unlike DPO or PPO). One core question is then: can we simply use KTO instead of 'ternarizing DPO'? What would be the benefits of TODO versus KTO?
(iii) The authors very vaguely comment in lines 122-123 that previous preference alignment studies "...overlook the potential of leveraging tied preference data to refine alignment, whereas our approach is complementary to most of them." It is unclear to which studies the proposed approach is complementary, and as mentioned above - it does not offer sufficient empirical validations to demonstrate that the approach is better than KTO, IPO, etc. I would suggest including direct comparisons to these other methods beyond DPO.

- All the experiments are based on the UltraFeedback dataset, and the training set size is limited to 20k data instances. The training set size might be a very important factor in defining the final performance (and benefits of ternary vs binary vs unary modeling as with KTO). Therefore, the paper should investigate how the standard DPO and KTO behave in comparison to TODO also for larger datasets (e.g., using the full UltraFeedback). Ideally, the work should experiment with another preference alignment dataset as well beyond UF, I would strongly suggest running experiments with another dataset, and compare against methods such as DPO and KTO also on larger datasets.

I am willing to reconsider my initial assessment and position if these central weaknesses get addressed by the authors.

**Questions:**

Q1 (related to the first, main point listed under "Weaknesses"). Speaking of previous work, how does your tie-aware BT model differ from or improve upon existing multi-class generalizations of the BT model that I provided under weaknesses plus the ones that you also cited? Are there any components, ideas or other aspects of prior approaches to BT generalisation that could be incorporated into or compared against your method?

Q2. It seems that the results are also dependent on the tie data ratio in the full training set of 20k instances. How can a practitioner determine the best (or task-optimal) choice of the tie data ratio? What if tie data is lacking for some specific use-cases?

Q3. How were hyperparameters from Appendix A.9 selected? Is there any substantial variation of the results based on the hyper-parameters?

---

> ### Author Response · Authors · 2024-11-24
> **Response to Reviewer iuqB[1/5]**
>
> Thank you for your valuable feedback. We would like to address each of your points individually.
>
>
> **W1-1: One of the core issues with the work lies in the fact that other researchers in `less recent times' have already attempted and derived generalised versions of the BT model, which better align with multi-class (i.e., non-binary) problems**
>
> - Thank you for bringing these three papers to our attention. They all address multi-class classification, where a ternary prediction is made for a given input feature, which differs from our focus on preference alignment, where a model predicts preference order given at least two inputs. Specifically:
>
>     - The first paper examines classification by estimating individual skills through paired team comparisons and proposes an iterative algorithm with theoretical guarantees. This is more aligned with reward model training rather than LLM alignment. Ties are treated as special cases (Section 7.3) using the ternary model from Rao & Kupper [2]. Our tie-aware BT model is also inspired by this 1967 paper, as noted in our manuscript. However, due to differences in objective functions, their algorithm and analysis do not extend to our case.
>     - The second paper is a preliminary conference version of the first, with nearly identical content.
>     - The third paper addresses multi-class classification using AdaBoost and proposes a loss function for ternary labels (Equation 9), which doesn't directly relate to our preference alignment or ternary probability model. While extending this loss function to a probability model is possible, it would require significant theoretical work and is beyond the scope of this paper.
>
> We will further discuss the differences between our work and related studies in the next response.

---

> ### Author Response · Authors · 2024-11-24
> **Response to Reviewer iuqB[2/5]**
>
> **W1-2: I would require the authors to provide a thorough overview of that relevant work and discuss why a new (tie-aware) derivation might be needed here, and whether previous solutions could be repurposed for the extension of the original, binary BT model**
>
> - Numerous studies have extended the binary BT model to handle ties, each differing primarily in their assumptions about the output score $s$ for an individual with strength $\lambda$. The earliest work by Glenn & David [1] is based on the Thurstone-Mosteller model, assuming that the score difference $s_1-s_2$ follows a Gaussian distribution with mean $\lambda_1-\lambda_2$ and unit variance. In contrast, the BT model assumes a logistic distribution for $s_1-s_2$. Rao & Kupper [2] extended the BT model by assuming that small values of $\ln(\lambda_1)-\ln(\lambda_2)$ result in ties. Davidson [3] adopted the "choice axiom," proposing that the probability of a tie is proportional to the geometric mean of the probabilities of non-tie relations, which shares the same asymptotic efficiency as Rao & Kupper's model. Recently, Baker & Scarf [4] suggested that scores follow a geometric distribution with strength $\lambda\in(0,1)$, aligning with the rules of golf. The table below summarizes these tie-aware models.
>
>
>
>     **Table R6: Comparison of different tie-aware ranking models. $\lambda_1$ and $\lambda_2$ represent the strengths of players 1 and 2, respectively. $\Delta=\lambda_1-\lambda_2$ and $F(\cdot)$ is the cumulative distribution function for the normal distribution. $r_{(12)}$ represents the probability of being tied.**
>
>     | Methods|Tie probability|Non-tie probability|
>     | -------------| --------- |  --------- |
>     |Glenn & David [1]|$r_{(12)}=F(\tau+\Delta)-F(-\tau+\Delta)$| $r_{12}=F(-\tau+\Delta)$|
>     |Rao & Kupper [2]|$r_{(12)}=\frac{(\phi^2-1)\lambda_1\lambda_2}{(\lambda_1+\phi\lambda_2)(\lambda_2+\phi\lambda1)}$|$r_{12}=\frac{\lambda_1}{\lambda_1+\phi\lambda_2}$|
>     |Davidson [3]| $r_{(12)}=\frac{\mu\sqrt{\lambda_1\lambda_2}}{\lambda_1+\lambda_2+\mu\sqrt{\lambda_1 \lambda_2}}$| $r_{12}=\frac{\lambda_1}{\lambda_1+\lambda_2+\mu\sqrt{\lambda_1 \lambda_2}}$|
>     |Baker & Scarf [4]|$r_{(12)}=\frac{1}{\lambda_1+\lambda_2+1}$|$r_{12}=\frac{\lambda_1}{\lambda_1+\lambda_2+1}$|
>
> - Our tie-aware BT model is not built from scratch but extends the model in Rao & Kupper [2], as explicitly mentioned. Firstly, we provide detailed derivation from the integral $r_{(12)}=\frac{1}{4}\int_{-(\ln\lambda_1-\ln\lambda_2)-\alpha}^{-(\ln\lambda_1-\ln\lambda_2)+\alpha} \text{sech}^2(y/2) dy$ to $r_{(12)}=\frac{(\phi^2-1)\lambda_1\lambda_2}{(\lambda_1+\phi\lambda_2)(\lambda_2+\phi\lambda1)}$. Secondly, we leverage the fact that it depends only on $r_1-r_2$ after introducing rewards $r_1,r_2$ and $\lambda_1=\exp{(r_1)}, \lambda_2=\exp{(r_2)}$, allowing integration into the DPO loss function. Lastly, Rao & Kupper proposes to use an iterative algorithm to obtain the maximum likelihood estimate of $\lambda_1,\lambda_2$ and $\phi$, which is not amenable in the presence of LLM. In our work, we empirically find that making $\phi$ trainable leads to poor performance and hence we set it as a hyper-parameter.
>
> - The model's reliance on $r_1-r_2$ rather than $r_1$ and $r_2$ individually is crucial, because the reward includes an untractable variable $Z(x)$ which can be eliminated in $r_1-r_2$ (please refer to Eq. (6) in our paper or Eqs. (4) and (5) in the DPO paper). In contrast, the models in Baker & Scarf [4] lack this property and cannot be adapted for DPO-style LLM alignments. While Davidson's [3] model has this property, it assumes the tie probability is proportional to the geometric mean of non-tie probabilities, which may not hold in real data. Glenn & David's [1] model shows potential but is not based on the BT model, and its effectiveness for LLM alignment remains unexplored.
>
> A discussion on other tie-extension models in alignment has been added in Section 7.

---

> ### Author Response · Authors · 2024-11-24
> **Response to Reviewer iuqB[3/5]**
>
> **Q1-1: Speaking of previous work, how does your tie-aware BT model differ from or improve upon existing multi-class generalizations of the BT model that I provided under weaknesses plus the ones that you also cited?**
>
> - Please refer to our previous replies of **W1-1 and W1-2**.
>
> **Q1-2: Are there any components, ideas or other aspects of prior approaches to BT generalisation that could be incorporated into or compared against your method?**
>
>
> - Please refer to our replies to **W1-1 and W1-2**. Although most of them cannot be directly incorporated, we can still learn some ideas. For example, the non-BT model in Glenn & David [1] was applied on football game, and it might also work on LLM alignment. Besides, the model in  Baker & Scarf [4] introduces no hyper-parameter, which simplifies LLM training if  can be adapted.
>
> **W2-1: Incorporating TOBT to other methods**
>
> - For RLHF-based methods, a straightforward integration way is to train a ternary reward model instead of the binary ones. The objective functions for the new reward model are composed of Eqs (12) and (13) in our manuscript. To this end, a ternary-labeled preference dataset is needed, which is not difficult to obtain based on common annotation methods.
>
> - For DPO-like methods, the integration varies by case. We take ODPO [5] as an example which adopts a loss function in the form of $-\log(\sigma(\mu - \Delta_{r}))$, where $\Delta_{r}$ introduces a preference margin depending on the ground-truth scores of the two inputs. By applying Eq. (15) in our manuscript, we can obtain a new loss incorporating tie data: $-\log\left(\frac{\exp(2\Delta_{r}-1)}{(1 + \exp(\mu + \Delta_{r}))(1 + \exp(\mu - \Delta_{r}))}\right)$, the derivation of which would be similar to TODO. For KTO, it seems difficult to find a trivial way to adapt to ternary preference decision, since it does not require paired preference data.
>
> - Due to page limitation and limited resources, we are not able to fully explore these ideas or do empirical experiments. We have added the discussion of potential integration of TOBT with other methods in Section 7.
>
> **W2-2:Comparison with other methods, especially KTO**
> - We compared TODO with recent methods such as SimPO, ODPO, and KTO on two preference datasets. The results, detailed in our **general response**, indicate that TODO outperforms other methods when tie data is present. Furthermore, TODO maintains competitive performance even when only binary preference data is available.
>
> - While KTO has the advantage of not requiring paired preference data, its performance was comparatively less competitive in our experiments. This may be due to a point noted in the KTO paper [6], which states, "...our theoretical analysis suggests that if your preference data has sufficiently little noise and sufficiently little intransitivity, then DPO will work better, as there is a risk of KTO underfitting"。
>
> - The discussion of TODO against other methods on both two dataset has been added in Section 7, the supporting experimental results have been added in Appendix A.13.
>
> **W2-3: TOBT complements binary BT alignment methods**
>
> - Thanks for the suggestion! We clarify as follows: by saying "our approach is complementary," we mean that the concept of LLMs learning from tie data can enhance existing methods, and our approach has the potential to be integrated with them. For further discussion, please refer to our response to **W2-1**.
>
> - We have revised the statements on the potential integration of TOBT with other methods in lines 122-123.

---

> ### Author Response · Authors · 2024-11-24
> **Response to Reviewer iuqB[4/5]**
>
> **W3-1:Rationale for selecting 20k data samples over larger sets**
>
>
> - Some researchers suggest that alignment methods may not require large training datasets, since improved performance is often attributed to response diversity [7-9]. To test this hypothesis, we conducted an experiment aligning Mistral-7B using 20k and 60k samples (nearly the entire Ultrafeedback dataset) with the same distribution. As shown in **Table R7,** MT Bench results indicate that the 20k dataset achieves competitive performance, even surpassing the 60k dataset. This validation result, along with our limited experimental resources, led us to focus subsequent experiments on the 20k sample size. Additionally, preference data is often scarce because collecting large amounts is costly and time-consuming. Therefore, training with fewer samples may align more closely with real-world scenarios.
>
>
>     **Table R7: MT Bench results on Mistral-7B aligned with 60k-size and 20k-size binary preference data using DPO (the scores are evaluated by gpt-4-turbo-2024-04-09)**
>
>     | **Turn**           | **Mistral-7B (SFT)** | **Mistral-7B+DPO (60k Ultrafeedback)** | **Mistral-7B+DPO (20k in the manuscript)** |
>     |------------------------------|-------------------------|-----------------------------|----------------------------|
>     | 1                   | 5.6875                 | 6.3375                     | 6.575                     |
>     | 2                   |5.4375                 | 6.475                      | 6.350                     |
>     | **Average Score**   | 5.56250                | 6.40625                    | **6.4625**
>
> **W3-2: Comparisons with other methods on additional datasets**
> As detailed in our **general response**, we have constructed a human-labeled dataset, Chatarena, which includes comparisons with KTO, SimPO, and ODPO. In scenarios involving tie data, TODO consistently achieved the highest test set accuracy and MT Bench scores across both datasets, as shown in **Tables R1 and R2**. In situations without tie data, TODO also demonstrated competitive performance.
>
> The discussion of TODO against other methods on both two dataset has been added in Section 7, the supporting experimental results have been added in Appendix A.13.
>
> **Q2: How can a practitioner determine the best (or task-optimal) choice of the tie data ratio? What if tie data is lacking for some specific use-cases**
>
> - While the optimal tie data ratio is hard to pinpoint, our empirical results suggest a range of 0.1–0.2. This range aligns closely with the natural frequency of tie data, which is approximately 0.17 in the UltraFeedback dataset. Within this range, TODO tends to outperform other binary BT model-based methods. However, as shown in **Figure 2**, excessive tie data can degrade performance.
>
> - In the absence of tie data, TODO can still be used for alignment, as detailed in Equation (18) of our paper. **Tables R1 and Table R2** demonstrate that TODO remains competitive even with a tie data ratio of 0.

---

> ### Author Response · Authors · 2024-11-24
> **Response to Reviewer iuqB[5/5]**
>
> **Q3: How were hyperparameters from Appendix A.9 selected? Is there any substantial variation of the results based on the hyper-parameters**
>
> - The selection of hyperparameters is based on previous works [8,10,11] to ensure consistency. The reference of hyperparameter settings is added in Section 5.2. Additionally, we conducted preliminary experiments with various hyperparameters before our main experiments. **Table R8** presents the results of two different hyperparameter settings on the MT Bench, demonstrating consistently that TODO outperforms DPO.
>
>     **Table R8: MT Bench scores on Llama 3-8B with different hyperparameters**
>
>
>     | Align       |  Hyperparametrs | **Tie ratio 0**   | **Tie ratio 0.1**  | **Tie ratio 0.2**  | **Tie ratio 0.3**  |
>     | ------------- | ---|------------- | ------------- | ------------- | ------------- |
>     | DPO  |Batch size 64, lr 5e-6 |6.74          | 6.53          | 6.05          | 6.13          |
>     | TODO   | Batch size 64, lr 5e-6|6.79     | 6.63      | 6.73      | 6.64     |
>     | DPO  |Batch size 128, lr 1e-6 (in the manuscript)|6.69         | 6.52          | 6.73         | 6.71         |
>     | TODO   | Batch size 128, lr 1e-6 (in the manuscript)|6.71      | 6.67     | 6.91      | 6.69      |
>
>
>
> #### **References**:
> [1] Glenn, W. A., and H. A. David. “Ties in Paired-Comparison Experiments Using a Modified Thurstone-Mosteller Model.”
>
> [2] Rao, P. V., and L. L. Kupper. “Ties in Paired-Comparison Experiments: A Generalization of the Bradley-Terry Model.”
>
> [3] Davidson, Roger R. “On Extending the Bradley-Terry Model to Accommodate Ties in Paired Comparison Experiments.”
>
> [4] Baker, Rose, and Philip Scarf. “Modifying Bradley–Terry and Other Ranking Models to Allow Ties.”
>
> [5] Amini, Afra, Tim Vieira, and Ryan Cotterell. “Direct Preference Optimization with an Offset.”
>
> [6] Ethayarajh, Kawin, Winnie Xu, Niklas Muennighoff, Dan Jurafsky, and Douwe Kiela. “KTO: Model Alignment as Prospect Theoretic Optimization.”
>
> [7] Nvidia, Bo Adler, Niket Agarwal, Ashwath Aithal, Dong H. Anh, Pallab Bhattacharya, Annika Brundyn, et al. “Nemotron-4 340B Technical Report.”
>
> [8] Saeidi, Amir, Shivanshu Verma, and Chitta Baral. “Insights into Alignment: Evaluating DPO and Its Variants Across Multiple Tasks.”
>
> [9] Song, Feifan, Bowen Yu, Hao Lang, Haiyang Yu, Fei Huang, Houfeng Wang, and Yongbin Li. “Scaling Data Diversity for Fine-Tuning Language Models in Human Alignment.”
>
> [10] Tunstall, Lewis, Edward Beeching, Nathan Lambert, Nazneen Rajani, Kashif Rasul, Younes Belkada, Shengyi Huang, et al. “Zephyr: Direct Distillation of LM Alignment.”
>
> [11] Meng, Yu, Mengzhou Xia, and Danqi Chen. “SimPO: Simple Preference Optimization with a Reference-Free Reward.”

---

> > ### Comment · Reviewer_iuqB · 2024-11-24
> > **Many thanks for the response!**
> >
> > I really appreciate the extent of responses, to me as well as to the other reviewers, and the answers did clarify some of the concerns I had with the first version - a proper discussion of Related Work and the latest set of results really puts things into perspective.
> >
> > I still have hesitations concerning at least two main points:
> > 1. Novelty of the idea - resolving ties is not new and similar principles have been proposed in prior work, they were just formalised for classification task rather than for (LLM) preference alignment.
> >
> > 2. Its practical usability still seems limited. While it definitely helps in certain setups with ties, I don't see practitioners switching from DPO to TODO given the current results. The gains over DPO are not so substantial - and it would be helpful to further understand when to use TODO and when to resort to DPO (and how to decide on that).

---

> ### Author Response · Authors · 2024-11-26
> **Thanks for your kind response**
>
> We are again very thankful for your time and support, and for raising the score. We have carefully considered your feedback and would like to address them below.
>
> **1. The novelty of our work.**
>
> We would like to emphasize that adapting existing multi-class classification models or tie-extension models to preference alignment is non-trivial, as noted in our responses to **W1-1 and W1-2**. For instance, the tie-extension method in **W1-1** based on AdaBoost is not a probabilistic model; the model in [4] relies on $r_1$ and $r_2$ individually rather than $r_1-r_2$, making it difficult to adapt to LLM alignment; and the model in [1] does not follow the BT model. Our work thoughtfully integrates tie-aware preference alignment in the context of LLMs.
>
> Furthermore, we have conducted extensive experiments to demonstrate the enhanced alignment capability of TODO with tie data, which cannot be captured by previous tie-aware classification tasks [1,3-4]. These results offer a fresh perspective on the construction and handling of preference data. Besides, we appreciate that both Reviewer **qtpx** and Reviewer **BkLA** have acknowledged the novelty of our tie-introduced alignment approach.
>
> **2. When to use TODO?**
>
> When only binary preference data is available, choosing between DPO and TODO depends on the "noise-to-signal" ratio of the data. TODO is preferable when noise levels are high, as it introduces a margin in the decision boundary that helps mitigate overfitting------a known issue with DPO. To validate this, we conducted an experiment by artificially injecting noise in binary preference data. Specifically, we randomly treat tie pairs as having clear preference differences. As shown in the following table, TODO achieved a higher MT Bench score than DPO in the noisy setting.
>
> **Table  : MT Bench scores with noise settings in DPO and TODO on Mistral-7B**
>
> | DPO on Ultrafeedback with noise |TODO on Ultrafeedback with noise |
> | ---------------- | ------------------------------ |
> |5.55                           | 5.64                           |
>
> When tie data is accessible, TODO is a better choice as demonstrated in our **general response**. In Section 6.3, we elucidate how TODO's superior  performance stems from its ability to embrace diversity of tied responses, which cannot be handled in DPO and its variants [5-7,13,14] or other binary model based methods [11,12].
>
> **3. The significance of improvement over DPO.**
>
> Thank you for your valuable feedback. In the challenging field of LLM alignment, even modest improvements are considered impactful due to the complexity of aligning LLM effectively. For example, recent works such as SimPO [11] (NeurIPS '24), MMPO [17] (EMNLP '24), and Curri-DPO [18] (EMNLP '24) have reported improvements on MT Bench over DPO ranging from 0 to 0.1, 0.01 to 0.13 (on Gemma models), and 0.13, respectively. These incremental gains have been recognized as significant advancements in the literature.
>
> In our study, as shown in Figure 2, TODO achieves improvements of 0.05 to 0.28 on the MT Bench with 0 and 0.1 ratios of tie data. These results indicate a more substantial enhancement compared to previous works, underscoring the effectiveness of our approach. Furthermore, Tables 2 and 3 demonstrate that TODO incurs a lower alignment tax than DPO. The alignment tax refers to the reduction in LLM performance on reasoning tasks that often accompanies alignment techniques [11,15,16]. TODO not only aligns the model more effectively but also preserves its reasoning capabilities.
>
> We believe these findings substantiate the significance of our improvements over DPO and contribute meaningful advancements to the field of LLM alignment.
>
> ### References:
>
> [12] Hong, Jiwoo, Noah Lee, and James Thorne. “ORPO: Monolithic Preference Optimization without Reference Model.”
>
> [13] Pal, Arka, Deep Karkhanis, Samuel Dooley, Manley Roberts, Siddartha Naidu, and Colin White. “Smaug: Fixing Failure Modes of Preference Optimisation with DPO-Positive.”
>
> [14] Xu, Haoran, Amr Sharaf, Yunmo Chen, Weiting Tan, Lingfeng Shen, Benjamin Van Durme, Kenton Murray, and Young Jin Kim. “Contrastive Preference Optimization: Pushing the Boundaries of LLM Performance in Machine Translation.”
>
> [15] Ouyang, Long, Jeff Wu, Xu Jiang, Diogo Almeida, Carroll L. Wainwright, Pamela Mishkin, Chong Zhang, et al. “Training Language Models to Follow Instructions with Human Feedback.”
>
> [16] Lin, Yong, Hangyu Lin, Wei Xiong, Shizhe Diao, Jianmeng Liu, Jipeng Zhang, Rui Pan, et al. “Mitigating the Alignment Tax of RLHF.”
>
> [17] Kim, Kyuyoung, Ah Jeong Seo, Hao Liu, Jinwoo Shin, and Kimin Lee. “Margin Matching Preference Optimization: Enhanced Model Alignment with Granular Feedback.”
>
> [18] Pattnaik, Pulkit, Rishabh Maheshwary, Kelechi Ogueji, Vikas Yadav, and Sathwik Tejaswi Madhusudhan. “Curry-DPO: Enhancing Alignment Using Curriculum Learning & Ranked Preferences.”

---

> > ### Comment · Reviewer_iuqB · 2024-11-27
> > **Thanks again...**
> >
> > For the detailed additional response.
> >
> > I really appreciate the amount of work invested into the response which clarified and tied a lot of loose ends with the original submission. I see a lot of merit in the work, but I am still not convinced, even after the additional empirical evidence, that the proposed method will be more valuable than DPO (and some other competitors) in general setups. Therefore, I'll stay with my current score.

---

> > > ### Author Response · Authors · 2024-12-02
> > > **Clarification of your concerns**
> > >
> > > Thank you for your continued feedback and for taking the time to evaluate our response. We believe that our conclusions are drawn from a widely recognized general setup in the literature, which we would like to summarize below.
> > >
> > > Firstly, our experiments were conducted on two popular LLMs, Mistral-7B and Llama 3-8B, using training hyperparameters consistent with established settings from prior research. The results consistently demonstrate the superiority of TODO over existing methods.
> > >
> > > Secondly, we performed a comprehensive comparison between TODO and DPO across various benchmarks, including in-distribution and out-distribution preference modeling accuracy, MT Bench scores, and a range of reasoning-intensive or knowledge-intensive tasks, utilizing both binary and ternary preference data. The results consistently show that the best outcomes are achieved when models are aligned with TODO using ternary preference data. Notably, TODO also outperforms DPO when using only binary data.
> > >
> > > Thirdly, we incorporated an additional human-labeled Chatarena dataset to evaluate the generalizability of our approach. The evaluation results from MT Bench scores and preference modeling accuracy across two datasets consistently demonstrate TOOD's superior performance over DPO and other methods.
> > >
> > > Finally, our work highlights the potential benefits of high-quality tie pairs in LLM alignments, offering a fresh perspective on the construction and handling of preference data for the community.
> > >
> > > We are somewhat puzzled by the term "general setups" in your response. We believe that our evaluation across diverse models, training datasets, evaluation methods, and benchmarks clearly indicates the robustness and significance of our approach.
> > >
> > > We sincerely hope that we have resolved all your concerns. We are confident that our work makes a significant and well-substantiated contribution, and your updated evaluation would mean a great deal to us. We kindly request that you reconsider the score. Thank you once again for your valuable feedback and consideration.

---

### Official Review · Reviewer_qtpx · 2024-11-05

**Soundness:** 2
**Presentation:** 3
**Contribution:** 3
**Rating:** 5
**Confidence:** 5

**Summary:**

The authors present an interesting approach to enhancing the alignment of LLMs through the use of ternary preferences. A limitation of binary models is their inability to effectively address tie situations in preference data, where two responses may be equally preferred or indistinguishable in quality. To address this issue, the authors propose an extension of the BT model, termed the Tie-rank Oriented Bradley-Terry (TOBT) model, which accommodates ties in preference relations. Building on this framework, they introduce a novel alignment algorithm called Tie-rank Oriented Direct Preference Optimization (TODO), specifically designed to manage ternary preference data that includes "prefer," "tie," and "disprefer." They conduct experiments comparing their TODO algorithm with the DPO. The results demonstrate that TODO consistently outperforms DPO in modeling preferences across both in-distribution and out-of-distribution datasets, as well as on several benchmarks. The paper also provides an analysis of the key factors contributing to TODO's improved performance.

**Strengths:**

1. The paper introduces a novel extension to the Bradley-Terry model by incorporating ties through the Tie-rank Oriented Bradley-Terry (TOBT) model. This allows for more nuanced preference modeling in LLM alignment.

2. The theoretical foundations of the TOBT model and the TODO algorithm are well-developed. The authors provide detailed derivations and clarify how the tie parameter α is integrated into the model.

3. The paper is generally well-written and structured. The methodology, including the mathematical formulations and the proposed algorithm, is clearly presented. The inclusion of figures and tables aids in understanding the results.

**Weaknesses:**

1. The extension from the traditional Bradley-Terry (BT) model to the Tie-rank Oriented BT (TOBT) model is central to the proposed method. However, the paper provides limited theoretical justification for this extension. The choice of the parameter α in the TOBT model appears somewhat arbitrary, and while Appendix A.3 attempts to justify its value based on balancing loss values, there is no rigorous analysis or sensitivity study demonstrating how different values of α affect the model's performance. Additionally, the paper lacks a thorough exploration of the probabilistic interpretation and properties of the TOBT model compared to the original BT model. A deeper theoretical analysis would strengthen the foundation of the proposed method and clarify its advantages and limitations.

2. The experiments primarily focus on datasets derived from Ultrafeedback and a select few benchmarks. While Ultrafeedback is a valuable resource, it may not fully represent the diversity and complexity of real-world human preference data, especially in terms of the prevalence and nature of tie data. The paper does not evaluate the method on more varied datasets, such as those involving different languages, domains, or more complex tasks where tie situations might behave differently. This raises concerns about the generalizability of the proposed method to broader applications and different types of data.

3. The paper compares the proposed TODO method primarily against Direct Preference Optimization (DPO). However, there are other existing methods in preference alignment and RLHF that could serve as relevant baselines, such as PPO, RPO, IPO), and KTO. By not including these comparisons, it is difficult to assess how TODO stands relative to the other existing methods in preference alignment. Including a broader range of baselines would provide a more comprehensive evaluation and strengthen the empirical claims.

4. While the paper highlights the prevalence of tie data in preference datasets and proposes methods to utilize it, But ties may often result from annotator uncertainty, ambiguity in prompts or responses, or inconsistencies among annotators. By treating tie data as equally informative as clear preferences without accounting for potential annotation noise or bias, the method might be incorporating noisy signals into the model. A discussion on how to handle annotation noise, perhaps by incorporating methods to assess annotator reliability or confidence, would strengthen the approach. Additionally, exploring strategies to improve data quality or to model uncertainty explicitly could enhance the effectiveness of the proposed method.

**Questions:**

1. Can TODO be integrated with other alignment methods beyond DPO, such as reinforcement learning from human feedback (RLHF)? If so, how would it interface with existing reward models?

2. Are there any trade-offs in using the TOBT model compared to traditional binary preference models? For example, does incorporating tie data introduce additional computational complexity or affect convergence?

---

> ### Author Response · Authors · 2024-11-24
> **Response to Reviewer qtpx [1/2]**
>
> Thank you for your kind and constructive feedback. We have carefully considered your comments and addressed them in detail below:
>
> **W1-1. Analysis and sensitivity study of $\alpha$**
>
> - Theoretically, both the BT-model and our TOBT model assume that the reward difference $\mu=r(x, y_1)- r(x, y_2)$ follows a logistic distribution with unit variance. For the BT-model, the mean is 0, whereas for the TOBT model, the mean is $\alpha$>0. During the initial training phase of LLM, the magnitude of $\mu$ is typically small. Consequently, a large $\alpha$ can result in a very large loss ($\mathbb{E}\log\sigma(\mu-\alpha)$) and lead to gradient vanishing, where $\sigma$ is the sigmoid function. From this standpoint, having $\alpha<1$ is advantageous since the gradient of $\sigma$ is less than 0.2 for $\alpha>1$. However, $\alpha$ should not be too close to 0 because the weights of tie data are propotional to $\exp(2\alpha)-1$. This presents a clear trade-off.
>
> - Empirically, in addition to the analysis provided in Appendix A.3, we further evaluate the performance of TODO with varying $\alpha$ values on our preference test set and MT Bench. The results, shown in **Tables R3 and R4**, indicate that performance is relatively insensitive to changes in $\alpha$ within a reasonable range, such as $\alpha \in (0.1, 0.8)$. However, divergence can occur when $\alpha$ is too large, aligning with our theoretical analysis. This behavior is similar to learning rate tuning, which can also lead to divergence beyond certain thresholds.
>
> - The analysis and corresponding results have been included in Appendix A.3.
>
>     **Table R3: Test-set accuracy in Mistral-7B model with different $\alpha$ values in TODO**
>
>     | Methods                | Ultrafeedback (tie ratio 0)     | Ultrafeedback (tie ratio 0.2)      | Chatarena (tie ratio 0)  | Chatarena (tie ratio 0.17)  |
>     | ------------- | --------- | --------- | --------- | ---------- |
>     | TODO($\alpha=0.1$)     | 77.20      | 76.40      | 76.80      | **78.47**  |
>     | TODO($\alpha=0.2$)     | **77.67** | 76.40      | 77.00      | 77.93      |
>     | TODO($\alpha=0.5$) | 77.33     | **76.87** | **77.53** | 77.87      |
>     | TODO($\alpha=0.8$) | Diverge    | 75.47 | 77.27 | 76.40 |
>     |  TODO($\alpha=1.2$)| Diverge    | 75.33 | Diverge     | Diverge    |
>
>
>     **Table R4: MT Bench scores in Mistral-7B model with different $\alpha$ values in TODO**
>
>     | Methods                | Ultrafeedback (tie ratio 0)     | Ultrafeedback (tie ratio 0.2)      | Chatarena (tie ratio 0)  | Chatarena (tie ratio 0.17)  |
>     | ---------------- | -------- | --------- | --------- | ---------- |
>     | TODO($\alpha=0.1$)      | **5.85**     | **5.96**      | 5.52      | **5.83**   |
>     | TODO($\alpha=0.2$)        | 5.44     | 5.94      | 5.53      | 5.58       |
>     | TODO($\alpha=0.5$)      | 5.73     | 5.91      | **5.69**      | 5.78       |
>     | TODO($\alpha=0.8$)  | Diverge | 5.81 | 5.48 | 5.51 |
>     |TODO($\alpha=1.2$) | Diverge | 5.41 | Diverge | Diverge |
>
> **W1-2. Probabilistic Comparison of TOBT and the BT Model**
>
> - Both the TOBT and BT models follow a logistic distribution. For paired data with a clear preference, TOBT introduces a margin $\alpha$ to shift the decision boundary towards $\alpha$, as shown in Equation (8) ($\sigma(\mu-\alpha)$), whereas the original BT model uses $\sigma(\mu)$ as the chosen probability, where $\mu$ represents the difference in estimated rewards between two responses. This creates a margin in the decision boundary in TODO, which is reserved for tie data. The following **Table R5** summarizes the probability comparison of the two models for tie and non-tie data.
> **Table R5: Probability comparison of the two models for tie or non-tie pairwise data.**
>
>     | Methods                   | Tie probability    |   Non-tie probability  |
>     | -------------             | --------- |  --------- |
>     | BT model      | 0|$\frac{1}{1+\exp(r(y_2)-r(y_1))}$   | $r_{12}=F(-\tau+\Delta)$
>     | Our TOBT       |$\frac{\exp(2\alpha)-1}{(1+\exp(r(y_1)-r(y_2)+\alpha))(1+\exp(r(y_2)-r(y_1)+\alpha))}$    | $\frac{1}{1+\exp(r(y_2)-r(y_1)+\alpha)}$ |
>
> - We have included a concise version of the above discussion in Section 4.2.

---

> ### Author Response · Authors · 2024-11-24
> **Response to Reviewer qtpx [2/2]**
>
> **W2： More real-word human preference dataset evaluations**
>
> - Ultrafeedback is a widely used preference dataset in LLM alignment research which has diverse prompts from over 6 resources, as detailed in Appendix A.8. Nevertheless, we agree  that more varied datasets can better demonstrate the generalizability of our method. Therefore, we conduct experiments on another human-labeled dataset Chatarena.
>
>     The Chatarena dataset, sourced from [lmsys-chatbot_arena_conversations](https://huggingface.co/datasets/agie-ai/lmsys-chatbot_arena_conversations), captures the diversity and complexity of real-world human preferences across **96** different languages. It includes pairs with clear preference differences as well as ties. In our experiments, we used a training set of 20,000 pairs with tie data ratios of 0 and 0.17 (the natural tie ratio of this dataset) and a test set of 1,500 randomly selected samples.
>
> - Results are provided in our **general response** in **Table R1** and **Table R2**. Consistent with the results on Ultrafeedback, TODO achieves the best performance compared to baselines in the presence of tie data. In the absence of tie data, TODO still delivers competitive performance. We believe these results demonstrate the generalizability of our method.
>
>
>
>
> **W3：Comparison with other methods**
>
> - We compared our method with recent preference alignment approaches, including KTO, SimPO, and ODPO, across different preference datasets. The results are available in our **general response** in **Tables R1** and **R2**.
>
> - In both datasets, TODO consistently outperforms other baselines when tie data is included, highlighting its advantages. Additionally, TODO with tie data surpasses methods without tie data in most cases, demonstrating the effectiveness of incorporating tie data.
>
> - The discussion of TODO against other methods on both two dataset has been added in Section 7, the supporting experimental results have been added in Appendix A.13.
>
> **W4：Discussion on data noise**
> - The noise-to-signal ratio in tie data can vary based on the annotation method. For instance, in Chatbot Arena, where users choose the best response or indicate a tie between two model outputs, we consider tie data to be as informative as data with clear preferences. Conversely, if tie data is constructed by grouping samples of equal quality evaluated by different people, it is likely to contain more noise.
>
> - Incorporating annotator confidence or consistency into training, such as assigning lower weights to samples with low confidence, could enhance model performance. We see this as a promising area for future research.
>
> - We acknowledge the importance of improving data quality, which remains an open challenge that is beyond the scope of this paper.
>
> **Q1: Integration of TOBT into other methods**
>
> - For RLHF-based methods, the integration is straightforward. The key change is to train a **ternary** reward model instead of the binary ones. The objective functions for this new reward model are outlined in Eqs (12) and (13) of our manuscript. To this end, a ternary-labeled preference dataset is needed, which is not difficult to obtain based on commonly used annotation processes.
>
>
> - For DPO-like methods, integration varies by case. Taking ODPO [1] as an example, which adopts a loss function $-\log(\sigma(\mu - \Delta_{r}))$, where $\Delta_{r}$ introduces a preference margin depending on the ground-truth quality scores of the inputs. By applying Eq. (15) in our manuscript, we can obtain a new loss incorporating tie data: $-\log\left(\frac{\exp(2\Delta_{r})-1}{(1 + \exp(\mu + \Delta_{r}))(1 + \exp(-\mu +\Delta_{r}))}\right)$, the derivation of which would be similar to TODO. However, the convergence and effectiveness of the new method remain to be explored. It would be interesting to study these methods in future works.
>
> - We have added the discussion of potential integration of TOBT with other methods in Section 7.
>
> **Q2: Trade-offs using TOBT compared to binary preference models**
> - To accommodate tie data, TOBT introduces a tunable hyperparameter $\alpha$ compared with most binary preference models. If $\alpha$ is not chosen appropriately, it may lead to training instability, as mentioned in our response to **W1-1**. Apart from this issue, TODO does not add extra computational complexity or affect convergence.
>
> #### **References**:
>
> [1] Amini, Afra, Tim Vieira, and Ryan Cotterell. "Direct Preference Optimization with an Offset."

---

> > ### Author Response · Authors · 2024-12-02
> > **Last day reminder**
> >
> > Dear Reviewer qtpx,
> >
> > We genuinely thank you again for your time and efforts and your constructive comments. We are eager to engage in further discussions to see if our response solves your concerns.
> >
> > As the deadline for the discussion period is nearing, we would greatly appreciate it if you could kindly let us know whether there are any further questions. Thank you for your attention to our work.
> >
> > Kind regards,
> >
> > The Authors

---

> ### Author Response · Authors · 2024-11-27
> **A Kind Reminder**
>
> Dear Reviewer **qtpx** ,
>
> We sincerely thank you for the valuable feedback you have provided, which has greatly improved our work. As the discussion phase nears its end, we kindly encourage you to share any additional feedback or updated thoughts regarding our responses. Your insights would greatly help us address any remaining concerns and further improve our work. We deeply appreciate your time and efforts.
>
> Kind regards,
>
> The Authors

---

### Author Response · Authors · 2024-11-24
**General  Response**

We sincerely appreciate the time and effort all reviewers have dedicated to evaluating our paper. We are grateful for the positive feedback, noting our work as "well-motivated", with "convincing results", "solid intuition", and being "well-written and structured". We also thank you for the constructive feedback and suggestions that have helped enhance our paper's quality.

As suggested by reviewers, we provide comparisons of our TODO with existing works and evaluate it on different datasets to demonstrate its generalization capabilities. We compare the proposed TODO method with KTO [1] (@reviewer **iuqB**), SimPO [2] (@reviewer **CZTq**), and ODPO [3] (@reviewer **BkLA**) on Mistral-7B, evaluated using MT Bench and our preference test sets (@reviewer **qtpx**). Additionally, we construct another preference dataset Chatarena [4] (@reviewer **qtpx**, **iuqB**), from diverse human-labeled open data. The new results are presented in **Tables R1 and R2**, with MT Bench scores evaluated by gpt-4o-2024-05-13. The hyperparameters used in these methods follow previous works [2,3]. For ODPO, distinct quality scores are required for the chosen and rejected samples in a pair, a condition only met in the Ultrafeedback dataset without tie data.


**Table R1: Accuracy on our preference testset**

| Methods                | Ultrafeedback (tie ratio 0)     | Ultrafeedback (tie ratio 0.2)      | Chatarena (tie ratio 0)  | Chatarena (tie ratio 0.17)  |
| ---------------------- | --------- | --------- | --------- | ---------- |
| DPO                    | 76.40     | 75.00     | **77.47**     | 76.33      |
| KTO                   | 73.80     | 74.27     | 74.80     | 74.80      |
| SimPO                  | **77.80** | 76.35     | 76.27     | 76.33      |
| ODPO                  | 76.78     | /         | /    | /          |
| **TODO**                  | 77.20       | **76.40**     | 76.80  | **78.47**       |



**Table R2: Results on MT Bench**

| Methods                | Ultrafeedback (tie ratio 0)     | Ultrafeedback (tie ratio 0.2)      | Chatarena (tie ratio 0)  | Chatarena (tie ratio 0.17)  |
| ---------------- | -------- | --------- | --------- | ---------- |
| DPO             | 5.84     | 5.55      | 5.50       | 5.71       |
| KTO            | 5.63     | 5.61      | 5.47      | 5.53       |
| SimPO           | 5.56     | 5.69      | 5.21      | 5.28       |
| ODPO            | **5.95** | /         | /  | /          |
| **TODO**       | 5.85     | **5.96**      | **5.52**     | **5.83**      |


We hope that our point-by-point responses below will effectively address your concerns.

#### **References:**

[1]Ethayarajh, Kawin, Winnie Xu, Niklas Muennighoff, Dan Jurafsky, and Douwe Kiela. “KTO: Model Alignment as Prospect Theoretic Optimization.”

[2]Meng, Yu, Mengzhou Xia, and Danqi Chen. “SimPO: Simple Preference Optimization with a Reference-Free Reward.”

[3]Amini, Afra, Tim Vieira, and Ryan Cotterell. “Direct Preference Optimization with an Offset.”

[4]https://huggingface.co/datasets/agie-ai/lmsys-chatbot_arena_conversations

---

### Meta-Review · Area_Chair_uuye · 2024-12-15

**Metareview:**

The authors introduce an extension of the Bradley-Terry (BT) model, called the Tie-rank Oriented Bradley-Terry (TOBT) model. The TOBT model incorporates ties, allowing for a more nuanced representation of preferences. Building on this foundation, the Tie-rank Oriented Direct Preference Optimization (TODO) algorithm is proposed to improve alignment further.

All reviewers found the paper well-written, the results reasonable, and the motivation clear. For the camera-ready version, experiments added during the rebuttal phase could be moved from the appendix to the main body to convince readers better. Additionally, reporting the sensitivity of hyperparameters in the main text would enhance usability and provide clearer guidance.

**Additional Comments On Reviewer Discussion:**

Reviewer CZTq raised concerns regarding the limited baselines, modest improvements, and reliance on additional annotation resources. In response, the authors conducted experiments to demonstrate that their method offers stable training performance across different hyperparameters and datasets. The authors also clarified that their tie data does not rely on additional annotations but rather maximizes the utilization of the existing annotations in the preference dataset.

Reviewer iuqB expressed concerns about the novelty of the idea, the practical usability of the method, and the limited improvements observed. The authors, however, argued that the observed improvements, though modest, are significant. On this point, I side with the reviewer, as the evaluation metrics (e.g., MT-Bench) are subject to fluctuations due to sampling variability.

Reviewer qtpx highlighted issues related to limited baselines, a constrained training dataset, and insufficient theoretical justification. The authors addressed these concerns during the rebuttal process by providing corresponding responses.

---

### Decision · Program_Chairs · 2025-01-22

Accept (Poster)